# DAGMA: Learning DAGs via M-matrices and a Log-Determinant Acyclicity Characterization

**Kevin Bello**[†‡]     **Bryon Aragam**[†]     **Pradeep Ravikumar**[‡]

[†]Booth School of Business, University of Chicago, Chicago, IL 60637
[‡]Machine Learning Department, Carnegie Mellon University, Pittsburgh, PA 15213

## Abstract

The combinatorial problem of learning directed acyclic graphs (DAGs) from data was recently framed as a purely continuous optimization problem by leveraging a differentiable acyclicity characterization of DAGs based on the trace of a matrix exponential function. Existing acyclicity characterizations are based on the idea that powers of an adjacency matrix contain information about walks and cycles. In this work, we propose a new acyclicity characterization based on the log-determinant (log-det) function, which leverages the nilpotency property of DAGs. To deal with the inherent asymmetries of a DAG, we relate the domain of our log-det characterization to the set of *M-matrices*, which is a key difference to the classical log-det function defined over the cone of positive definite matrices. Similar to acyclicity functions previously proposed, our characterization is also exact and differentiable. However, when compared to existing characterizations, our log-det function: (1) Is better at detecting large cycles; (2) Has better-behaved gradients; and (3) Its runtime is in practice about an order of magnitude faster. From the optimization side, we drop the typically used augmented Lagrangian scheme and propose DAGMA (*Directed Acyclic Graphs via M-matrices for Acyclicity*), a method that resembles the central path for barrier methods. Each point in the central path of DAGMA is a solution to an unconstrained problem regularized by our log-det function, then we show that at the limit of the central path the solution is guaranteed to be a DAG. Finally, we provide extensive experiments for *linear* and *nonlinear* SEMs and show that our approach can reach large speed-ups and smaller structural Hamming distances against state-of-the-art methods. Code implementing the proposed method is open-source and publicly available at https://github.com/kevinsbello/dagma.

## 1 Introduction

Structural equation models (SEMs) [44] are a standard modeling tool in several fields such as economics, social sciences, genetics, and causal inference, to name a few. Under this framework, in its general form, the value of each variable in the model is assigned by a general nonlinear, nonparametric function that takes as input the values of other variables in the model, thus, every SEM can be associated to a graphical model. In particular, we will consider graphical models that are directed acyclic graphs (DAGs).

A long-standing and active research area deals with the problem of learning the graphical structure (DAG) given passively observed data (a.k.a. causal discovery). Computationally, this problem is well-known to be NP-hard in general [10, 12], mainly due to the combinatorial nature of the space of DAGs. In this work, we will follow a score-based approach, a popular learning framework where the goal is to find a DAG that minimizes a *given* score [21, 11]. Recently, Zheng et al. [58] proposed an exact smooth nonconvex characterization of acyclicity which opened the door to solving the originally

36th Conference on Neural Information Processing Systems (NeurIPS 2022).

combinatorial problem via a set of tools that work in the fully continuous regime (e.g., gradient-based methods).

Let $W \in \mathbb{R}^{d \times d}$ be a weighted adjacency matrix of a graph $G$ of $d$ nodes, and let $W \circ W$ denote the Hadamard product. The acyclicity function introduced by Zheng et al. [58] is defined as $h_{\mathrm{expm}}(W) = \mathrm{Tr}(e^{W \circ W}) - d$, where $\mathrm{Tr}$ denotes the trace of a matrix, and it was shown that $h_{\mathrm{expm}}(W) = 0$ if and only if $W$ corresponds to a DAG. A follow-up work [56] proposed another acyclicity function, which can be computed slightly faster, defined as $h_{\mathrm{poly}}(W) = \mathrm{Tr}((I + \frac{1}{d} W \circ W)^d) - d$, where $I$ is the identity matrix. It was similarly shown by Yu et al. [56] that $h_{\mathrm{poly}}(W) = 0$ if and only if $W$ corresponds to a DAG. While seemingly different, both $h_{\mathrm{expm}}$ and $h_{\mathrm{poly}}$ are functions of the form $\mathrm{Tr}(\sum_{k=0}^{d} c_p (W \circ W)^k) - d$ for some $c_k > 0$, as noted by Wei et al. [55]. To the best of our knowledge, all subsequent work [e.g., 59, 26, 60, 37, 38, 36, 25, 41, to name a few] that has built upon the idea of using a continuous acyclicity characterization have used either $h_{\mathrm{expm}}$ or $h_{\mathrm{poly}}$, or some acyclicity characterization in the form of a trace of a sum of matrix powers. The latter should come as no surprise, after all, a nonzero diagonal entry of the matrix power $(W \circ W)^k$ reveals the existence of a closed walk of length $k$ in $W$.

**Contributions.** In this work, we propose a new acyclicity function that, as $h_{\mathrm{expm}}$ and $h_{\mathrm{poly}}$, is both an exact and a smooth acyclicity characterization but that also possesses several advantages when compared to $h_{\mathrm{expm}}$ and $h_{\mathrm{poly}}$. Specifically, we make the following set of contributions:

1. We propose a novel acyclicity characterization based on the log-determinant (log-det) function (see Theorem 1 and Section 3). In contrast to the classical log-det function defined over the cone of positive definite matrices, we define the domain of our log-det function to be the set of M-matrices due to the inherent asymmetries of DAGs. To our knowledge, we are the first to connect the notion of M-matrices to acyclicity and structure learning for DAGs.

2. We provide a detailed study of the properties of our log-det characterization in Section 3.1. First, we establish the similarities of our log-det function to other existing functions such as $h_{\mathrm{expm}}$ and $h_{\mathrm{poly}}$. Second, we formally argue why these functions can be regarded as acyclicity regularizers, similar in spirit to the classical $\ell_1$ and $\ell_2$ regularizers. Third, we show that our log-det function is an *invex* function, i.e., all stationary points are global minimum, moreover, these stationary points correspond to DAGs.

3. In Section 3.2, we present three arguments as to why our log-det characterization could be preferred over other existing acyclicity functions. Briefly, our log-det function is better at detecting large cycles, has better behaved gradients, and can be computed in about an order of magnitude faster than $h_{\mathrm{expm}}$ and $h_{\mathrm{poly}}$.

4. Motivated by the properties of our log-det function, in Section 4, we present DAGMA (*Directed Acyclic Graphs via M-matrices for Acyclicity*), a method that resembles the widely known central path approach for barrier methods [39]. We show that, at the limit of the central path, the solution is guaranteed to be a DAG. In contrast to the commonly adopted augmented Lagrangian scheme (originally proposed in [58]) each point in the central path of DAGMA is a solution to an unconstrained problem regularized by our log-det function.

5. Finally, in Section 5 and Appendix C, we provide extensive experiments for *linear* and *nonlinear* SEMs under different score functions (both least squares and log-likelihood), where we show that DAGMA is capable of obtaining DAGs with *better accuracy*, i.e., lower structural Hamming distance (SHD), in a *much faster* way than the state-of-the-art.

## 1.1  Related work

The vast majority of methods for learning DAGs can be categorized into two groups: constraint-based algorithms, which rely on conditional independence tests; and score-based algorithms, which focus on finding a DAG that minimizes a given score/loss function. We briefly mention classical constraint-based methods as we follow a score-based approach. [50] developed the PC algorithm, a popular general method that learns the Markov equivalence class. Other algorithms such as [52, 29] are based on local Markov boundary search. Finally, hybrid approaches that combine constraint-based learning with score-based learning, such as [53, 15].

In the line of score-based methods, popular score functions include BDeu [21], BIC [31], and MDL [7]. Works that study linear Gaussian SEMs include [2, 3, 16, 17, 32, 45], and for linear non-Gaussian SEMs [27, 49]. For nonlinear SEMs, we note works on additive models [9, 13, 54], additive noise models [23, 46, 35], generalized linear models [43, 42, 19], and general nonlinear SEMs [34, 18].

More closely related to our work is the line of work built on the nonconvex continuous framework of Zheng et al. [58], such as, [59, 26, 60, 37, 36, 25, 41]. In contrast to our work, all of the aforementioned methods rely on the nonconvex acyclicity functions $h_{\mathrm{expm}}$ or $h_{\mathrm{poly}}$, and with the exception of [37], all of these works also use the augmented Lagrangian scheme. Finally, the NoCurl method [57] also departs from using $h_{\mathrm{expm}}$, although no other acyclicity constraint is proposed. Two immediate distinctions can be made to our work. First, we propose a novel acyclicity function based on the log-det function which we show to be preferable to $h_{\mathrm{expm}}$ and $h_{\mathrm{poly}}$. Second, we drop the commonly adopted augmented Lagrangian scheme to solve the constrained problem and instead follow a central path approach to leverage the barrier property of our log-det function.

**Remark 1.** *To avoid confusion, we also note that in the GOLEM method of Ng et al. [37] the score includes a log-determinant function of the form* $\log|\det(I - W)|$ *which stems from the* Gaussian *log-likelihood. While this expression is zero if $W$ corresponds to a DAG, it is **not** an exact acyclicity characterization (i.e.* $\log|\det(I - W)| = 0$ *does not imply $W$ is a DAG). By contrast, the use of* M-*matrices in our work is crucial to translating the log-det function into a valid acyclicity regularizer. Moreover, it is not obvious how to extend GOLEM to arbitrary score functions, as their analysis is specific to the Gaussian likelihood function.*

## 2 Notation and background

**Notation.** We use $[d]$ to denote the set of integers $\{1 \ldots d\}$. For a square matrix $A$, we use $\lambda_i(A)$ to denote its $i$-th minimum eigenvalue, and use $\rho(A)$ to denote its spectral radius. Also, we use $\mathrm{Tr}(A)$, and $\det(A)$ to denote the trace and determinant of $A$. For matrices $A, B$, we let $A \circ B$ represent the element-wise or Hadamard product, moreover, the expression $A \geq B$ is entrywise, i.e., $A_{i,j} \geq B_{i,j}$. Then, we say that a matrix $A$ is nonnegative whenever $A \geq 0$. For a complex number $a + bi$, we let $\Re(a + bi) = a$ denote its real part. We use $\|\cdot\|_p$ to denote the vector $\ell_p$-norm, and $\|\cdot\|_{L^p}$ is the $L^p$-norm on functions. Lastly, $i \to j$ and $i \rightsquigarrow j$ represent an edge from $i$ to $j$ and a directed walk from $i$ to $j$, respectively.

Let $X = (X_1, \ldots, X_d)$ be a $d$-dimensional random vector. In its general form, a (nonparametric) structural equation model (SEM) consists of a set of equations of the form:

$$X_j = f_j(X, Z_j), \ \forall j \in [d], \tag{1}$$

where each $f_j : \mathbb{R}^{d+1} \to \mathbb{R}$ is a nonlinear nonparametric function, and $Z_j$ is an exogenous variable representing errors due to omitted factors. We consider the Markovian model, which assumes that each $Z_j$ is an independent random variable. Note that each $f_j$ depends only on a subset of $X$ (i.e., the parents of $X_j$) and $Z_j$; nonetheless, to simplify notation we ensure that each $f_j$ is defined on the same space. Then $f = (f_1, \ldots, f_d)$ induces a graphical structure, where we focus on directed acyclic graphs.

For any joint distribution over $Z = (Z_1, \ldots, Z_d)$, the functions $f_j$ define a joint distribution $\mathbb{P}(X)$ over the observed data, and a graph $G(f)$ via the dependencies in each $f_j$. Then, our goal is to learn $G(f)$ given $n$ i.i.d. samples from $\mathbb{P}(X)$. In score-based learning, given a data matrix $\boldsymbol{X} = [\boldsymbol{x}_1, \ldots, \boldsymbol{x}_d] \in \mathbb{R}^{n \times d}$, we define a score function $Q(f; \boldsymbol{X})$ to measure the 'quality' of a candidate SEM as follows: $Q(f; \boldsymbol{X}) = \sum_{j=1}^d \mathsf{loss}(\boldsymbol{x}_j, f_j(\boldsymbol{X}))$, where we adopt the convention that $f_j(\boldsymbol{X}) \in \mathbb{R}^n$. Here $\mathsf{loss}$ can be any loss function such as least squares $\mathsf{loss}(\boldsymbol{u}, \boldsymbol{v}) = \frac{1}{n}\sum_{i=1}^n (u_i - v_i)^2$ or the log-likelihood function, often augmented with a penalty such as BIC or $\ell_1$. Given the score function $Q$ and a family of functions $\mathcal{F}$, we seek to find the $f \in \mathcal{F}$ that minimizes the score, i.e.,

$$\min_{f \in \mathcal{F}} Q(f; \boldsymbol{X}) \quad \text{subject to} \quad G(f) \in \mathsf{DAGs}. \tag{2}$$

Similar to [59], we consider that each $f_j$ lives in a Sobolev space of square-integrable functions whose derivatives are also square integrable. Then, let $\partial_k f_j$ denote the partial derivative of $f_j$ w.r.t. $X_k$, it is easy to see that $f_j$ is independent of $X_k$ if and only if $\|\partial_k f_j\|_{L^2} = 0$. With this observation, we construct the matrix $W(f) \in \mathbb{R}^{d \times d}$ with entries $[W(f)]_{i,j} \overset{\text{def}}{=} \|\partial_i f_j\|_{L^2}$, which precisely encodes

the graphical structure amongst the variables $X_j$. That is $G(f) \in \mathsf{DAGs} \iff W(f) \in \mathsf{DAGs}$, where $W$ is interpreted as the usual weighted adjacency matrix.

In practice, $f$ is replaced with a flexible family of parametrized functions such as deep neural networks, so that problem (2) is finite dimensional. Finally, note that model (1) includes several models as special cases, e.g., additive noise models, generalized linear models, additive models, polynomial regression, and index models. Previous work has studied the identifiability of several of these models, e.g., [23, 49, 46, 43, 42, 27]. In the sequel, we assume that the model is chosen such that the graph $G(f)$ is uniquely defined from (2).

## 3 A new characterization of acyclicity via log-determinant and M-matrices

In this section, we present our acyclicity characterization and study its properties. To declutter notation, in this section we simply write $W$ instead of $W(f)$ to denote the *weighted* adjacency matrix of a graph; however, it should be clear that $W$ depends on functions $f_j$ as explained in the previous section.

We develop our characterization by first noting that for any *nonnegative* weighted adjacency matrix $W$, we have that $W \in \mathsf{DAGs}$ if and only if $W$ is a nilpotent matrix, or equivalently, all the eigenvalues of $W$ are zero, i.e., $\lambda_i(W) = 0, \forall i \in [d]$. Then, for any $W \in \mathbb{R}^{d \times d}$, we have the following obvious implications:

$$W \in \mathsf{DAGs} \iff (W \circ W) \in \mathsf{DAGs} \iff s - \lambda_i(W \circ W) = s, \forall i \in [d], \forall s \in \mathbb{R} \quad (3)$$

$$\implies \prod_{i=1}^{d} s - \lambda_i(W \circ W) = \det(sI - W \circ W) = s^d. \quad (4)$$

Implication (4) can be thought of as a relaxation of acyclicity in the sense that all DAGs satisfy (4), but not all $W$ that satisfy (4) are DAGs. For example, let $s = 1$ and $W \circ W = \left[\begin{smallmatrix} 2 & 0 \\ 0 & 2 \end{smallmatrix}\right]$, then it is clear that $\det(sI - W \circ W) = 1$ and, thus, (4) is satisfied; however, clearly $W$ is not a DAG.

Thus, two immediate questions arise: (i) *Does there exist a domain for $W$ such that* (4) $\implies$ (3)*?* (ii) *If so, what is the description of such domain?* We answer (i) in the affirmative, and answer (ii) by relating the domain of $W$ to the set of M-matrices, which is defined below.

**Definition 1** (M-matrix[1]). *An M-matrix is a matrix $A \in \mathbb{R}^{d \times d}$ of the form $A = sI - B$, where $B \geq 0$ and $s > \rho(B)$.*

M-matrices were introduced by Ostrowski [40] and arise in a variety of areas including input-output analysis in economics, linear complementarity problems in operations research, finite difference methods for partial differential equations, and Markov chains in stochastic processes. To the best of our knowledge, we are the first to connect the notion of M-matrices to graphical model structure learning through an acyclicity characterization.

The following proposition is an immediate consequence of Definition 1:

**Proposition 1** (Berman and Plemmons [5]). *Let $A \in \mathbb{R}^{d \times d}$ be an M-matrix, then:*

*(i) $\Re(\lambda_i(A)) > 0$, for all $i \in [d]$.*      *(ii) $A^{-1}$ exists and is nonnegative, i.e., $A^{-1} \geq 0$.*

In the above, item (i) states that the eigenvalues of an M-matrix lie in the open right-half plane. Matrices which satisfy the latter property are also known as positive stable matrices. We thus have that M-matrices are special cases of positive stable matrices. It follows that the determinant of any M-matrix is positive.[2] This fact will be used for defining $h^s_{\mathrm{ldet}}(W)$, our acyclicity characterization given in Theorem 1. Finally, the nonnegativity of the inverse from item (ii) will be used to understand some properties of the gradient of $h^s_{\mathrm{ldet}}(W)$.

We now define the domain over which (4) $\implies$ (3) (see Theorem 1). For any $s > 0$, define

$$\mathbb{W}^s = \{W \in \mathbb{R}^{d \times d} \mid s > \rho(W \circ W)\}, \quad (5)$$

---

[1]More precisely, we consider the definition of a non-singular M-matrix, which is sufficient for the purposes of this work.

[2]Note that due to asymmetries it is possible for an M-matrix to have complex eigenvalues. However, since we work with matrices with real entries, the complex eigenvalues come in conjugate pairs.

i.e., $\mathbb{W}^s$ is the set of real matrices whose entry-wise square given by $W \circ W$ have spectral radius less than $s$. The following lemma lists some relevant properties of $\mathbb{W}^s$.

**Lemma 1.** *Let $\mathbb{W}^s$ defined as in* (5). *Then, for all $s > 0$:*

    *(i)* DAGs $\subset \mathbb{W}^s$.        *(ii)* $\mathbb{W}^s$ *is path-connected.*       *(iii)* $\mathbb{W}^s \subset \mathbb{W}^t$ *for any $t > s$.*

In Lemma 1, item (i) implies that the $W$ we look for is in the interior of $\mathbb{W}^s$; item (ii) indicates that we can find a path from any point in $\mathbb{W}^s$ to any DAG without leaving the set $\mathbb{W}^s$; item (iii) shows that one can vary $s$ to enlarge or shrink the set $\mathbb{W}^s$.

Having defined the domain set $\mathbb{W}^s$, we now define our acyclicity characterization $h^s_{\text{ldet}} : \mathbb{W}^s \to \mathbb{R}$. Recall that item (i) in Proposition 1 implies that applying the logarithm function to the determinant of an M-matrix is always well defined, which motivates our following result.

**Theorem 1** (Log-determinant characterization). *Let $s > 0$ and let $h^s_{\text{ldet}} : \mathbb{W}^s \to \mathbb{R}$ be defined as* $h^s_{\text{ldet}}(W) \overset{\text{def}}{=} -\log \det(sI - W \circ W) + d \log s$. *Then, the following holds:*

    *(i)* $h^s_{\text{ldet}}(W) \geq 0$, *with $h^s_{\text{ldet}}(W) = 0$ if and only if $W$ is a DAG.*

    *(ii)* $\nabla h^s_{\text{ldet}}(W) = 2(sI - W \circ W)^{-\top} \circ W$, *with $\nabla h^s_{\text{ldet}}(W) = 0$ if and only if $W$ is a DAG.*

## 3.1 Properties of $h^s_{\text{ldet}}(W)$

In this section, we list several properties of our acyclicity characterization. The first property we discuss is related to the entries of $\nabla h^s_{\text{ldet}}(W)$.

**Lemma 2.** *For all $i, j \in [d]$, $[\nabla h^s_{\text{ldet}}(W)]_{i,j} = 0$ if and only if $W_{i,j} = 0$ or there is no directed walk from $j$ to $i$. Equivalently, $[\nabla h^s_{\text{ldet}}(W)]_{i,j} \neq 0$ if and only if the edge $i \to j$ is part of some cycle in $W$. Finally, whenever $[\nabla h^s_{\text{ldet}}(W)]_{i,j} \neq 0$, we have that $\text{sign}([\nabla h^s_{\text{ldet}}(W)]_{i,j}) = \text{sign}(W_{i,j})$.*

The lemma above characterizes the nonzero entries and their signs of the gradient of $h^s_{\text{ldet}}$. This property formally offers a *regularizer* perspective for $h^s_{\text{ldet}}$, which we highlight next.

**Remark 2** (A regularizer viewpoint). *The function $h^s_{\text{ldet}}$ promotes small parameters values in much the same way that the classical $\ell_1$ and $\ell_2$ regularizers do. In contrast to the latter regularizers, $h^s_{\text{ldet}}$ will only shrink the value of a parameter $W_{i,j}$ if and only if the edge $(i,j)$ is part of some cycle in $W$, as prescribed by Lemma 2.*

Recall that $h_{\text{expm}}(W) = \text{Tr}(e^{W \circ W}) - d$ and $h_{\text{poly}}(W) = \text{Tr}((I + \frac{1}{d}W \circ W)^d) - d$. It was noted by [55] that acyclicity characterizations of the form $\text{Tr}(\sum_{p=1}^d c_p(W \circ W)^d)$ for $c_p > 0$ also have the property in Lemma 2. This implies that $h_{\text{expm}}$ and $h_{\text{poly}}$ hold the property above and can also be interpreted as acyclicity regularizers. We note that the interesting part here is that $h^s_{\text{ldet}}$ holds this property besides being different in nature to $h_{\text{expm}}$ and $h_{\text{poly}}$.

Next, we state an important consequence of Lemma 2, which is related to the direction of $\nabla h^s_{\text{ldet}}(W)$.

**Corollary 1.** *At any $W \in \mathbb{W}^s$, the negative gradient $\nabla h^s_{\text{ldet}}(W)$ points towards the interior of $\mathbb{W}^s$.*

In optimization, the Hessian matrix plays an important role as it contains relevant information about saddle points and local extrema of a function, and is key to Newton-type methods. Another appealing property of $h^s_{\text{ldet}}(W)$ is that it has a Hessian described by a simple closed-form expression.

**Lemma 3.** *The Hessian of $h^s_{\text{ldet}}(W)$, which resides in $\mathbb{R}^{d^2 \times d^2}$, is given by:*

$$\nabla^2 h^s_{\text{ldet}}(W) = 4 \, \text{Diag}(\text{vec}(W))(N \otimes N^\top) \, \text{Diag}(\text{vec}(W^\top))K^{dd} + 2 \, \text{Diag}(\text{vec}(N^\top)),$$

*where $N = (sI - W \circ W)^{-1}$, $\otimes$ denotes the Kronecker product, and $K^{dd}$ is the $d^2 \times d^2$ commutation matrix such that $K^{dd} \text{vec}(A) = \text{vec}(A^\top)$, for any $d \times d$ matrix $A$.*

Here we note that among $h_{\text{expm}}, h_{\text{poly}}$ and $h^s_{\text{ldet}}$, only $h^s_{\text{ldet}}$ has a *tractable* expression for the Hessian. Furthermore, note that $\nabla^2 h^s_{\text{ldet}}(W)$ is indexed by vertex pairs so that the entry $[\nabla^2 h^s_{\text{ldet}}(W)]_{(k,l),(p,q)}$ corresponds to the second partial derivative $\frac{\partial^2 h^s_{\text{ldet}}}{\partial W_{k,l} \partial W_{p,q}}$. Using Lemma 3, we can characterize the nonzero entries, and their signs, of the Hessian of $h^s_{\text{ldet}}$.

**Corollary 2.** *The entries of the Hessian $\nabla^2 h_{\mathrm{ldet}}^s(W)$ are described as follows:*

$$\left[\nabla^2 h_{\mathrm{ldet}}^s(W)\right]_{(k,l),(p,q)} = \begin{cases} 4W_{l,k}N_{k,q}N_{p,l}W_{q,p} & \text{if } (k,l) \neq (p,q), \\ 4(W_{l,k})^2(N_{k,l})^2 + 2N_{k,l} & \text{if } (k,l) = (p,q), \end{cases}$$

*where $N = (sI - W \circ W)^{-1}$. Moreover, an off-diagonal entry $[\nabla^2 h_{\mathrm{ldet}}^s(W)]_{(k,l),(p,q)}$ is nonzero if and only if there exists a cycle in $W$ of the form $q \to p \rightsquigarrow l \to k \rightsquigarrow q$, and has a sign equal to $\mathrm{sign}(W_{l,k}W_{q,p})$. Lastly, a diagonal entry $[\nabla^2 h_{\mathrm{ldet}}^s(W)]_{(k,l),(k,l)}$ is nonzero if and only if there exists a directed walk from $k$ to $l$, and its sign is always positive.*

Recall from Theorem 1 that all DAGs attain the minimum value and are critical points of $h_{\mathrm{ldet}}^s$; thus, DAGs are local (and global) minimum of $h_{\mathrm{ldet}}^s$ and the Hessian matrix evaluated at a DAG must be positive (semi)definite. Let us corroborate the latter, when $W$ is a DAG, from Corollary 2 we have that the off-diagonal elements of $\nabla^2 h_{\mathrm{ldet}}^s(W)$ are zero, while the diagonal entries are nonnegative. That is, the Hessian is positive semidefinite whenever $W$ is a DAG. This implies that all its stationary points are global minima: Such functions are called *invex* [20, 30].

**Corollary 3.** *Let $s > 0$. Then, $h_{\mathrm{ldet}}^s(W)$ is an invex function, i.e., all its stationary points are global minima, and these correspond to DAGs.*

We note that even though $h_{\mathrm{expm}}$ and $h_{\mathrm{poly}}$ are also invex functions, they were not explicitly considered as invex functions before. In fact, it was noted in [58] that DAGs were global minima of $h_{\mathrm{expm}}$ but no characterization of its stationary points were given. Wei et al. [55] noted that DAGs were stationary points of $h_{\mathrm{expm}}$ and $h_{\mathrm{poly}}$ but the notion of invexity was not explicitly stated.

**Remark 3** (A dynamical system perspective). *The importance of invexity here is that in the eyes of $h_{\mathrm{ldet}}^s$, all DAGs are the same. That is, DAGs correspond to the set of attractors in $h_{\mathrm{ldet}}^s$, and depending on the initial condition, the system will converge to a different attractor. This offers the following viewpoint for the role of the score function $Q(f; \mathbf{X})$, namely, "use the score $Q$ to find a basin of attraction such that the force field of $h_{\mathrm{ldet}}^s$ will dictate the trajectory towards a DAG that is equal or close to the ground-truth".*

In Figure 1, we illustrate in a toy example the properties discussed in this subsection.

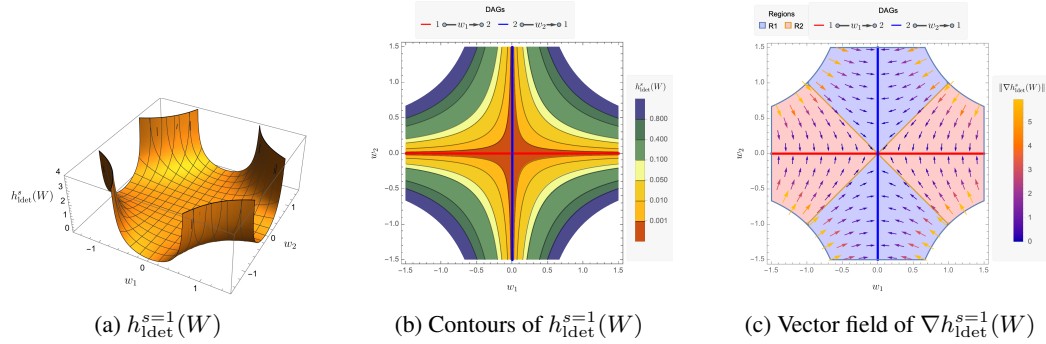

(a) $h_{\mathrm{ldet}}^{s=1}(W)$      (b) Contours of $h_{\mathrm{ldet}}^{s=1}(W)$      (c) Vector field of $\nabla h_{\mathrm{ldet}}^{s=1}(W)$

Figure 1: Behavior of $h_{\mathrm{ldet}}^s$ for $W = \begin{bmatrix} 0 & w_1 \\ w_2 & 0 \end{bmatrix}$. Here clearly $W$ is a DAG whenever one of $w_1$ or $w_2$ (or both) are zero. In particular, for (c) we note the perspective given in Remark 3, i.e., starting at any point in region R2 will converge to attractors (DAGs) of the form $X_1 \to X_2$ (red line); while starting at any point in region R1 will converge to attractors (DAGs) of the form $X_2 \to X_1$ (blue line).

## 3.2 Why the log-determinant regularizer is preferable to existing acyclicity regularizers

In this section, we present three arguments as to why one should use $h_{\mathrm{ldet}}^s$ instead of existing functions such as $h_{\mathrm{expm}}$ and $h_{\mathrm{poly}}$. We invite the reader to look at Appendix B for additional details.

**Argument (i).** $h_{\mathrm{ldet}}^s$ **does not diminish cycles of any length.** Let us expand the functions $h_{\mathrm{expm}}$ and $h_{\mathrm{poly}}$ in their sum of matrix powers form, that is, $h_{\mathrm{expm}}(W) = \sum_{k=0}^{\infty} 1/k! \, \mathrm{Tr}((W \circ W)^k) - d$ and $h_{\mathrm{poly}}(W) = \sum_{k=0}^{d} \binom{d}{k}/d^k \, \mathrm{Tr}((W \circ W)^k) - d$. Recall also that the entry $[(W \circ W)^k]_{i,i}$ represents

the sum of weighted walks from node $i$ to node $i$ of length $k$, where each edge has weight $w_{u,v}^2$. Thus, one can notice that if $W$ has cycles of length $k$, their contribution to $h_{\mathrm{expm}}$ and $h_{\mathrm{poly}}$ are *diminished* by $1/k!$ and $\binom{d}{k}/d^k$, respectively. Numerically, the latter can be problematic for the following reason: Cycles of length $k$ can go undetected even for small values of $d$ and $k$. In practice, a value of $h_{\mathrm{expm}}, h_{\mathrm{poly}} \in [10^{-8}, 10^{-5}]$ is typically regarded as zero [55, 58]. Consider a cycle graph of $d$ nodes where each edge weight is $+1$ or $-1$.[3] The plot in Figure 2 shows how the values of $h_{\mathrm{expm}}$ and $h_{\mathrm{poly}}$ decay much faster than that of $h_{\mathrm{ldet}}^s$; in fact, at $d = 13$ we already observe $h_{\mathrm{expm}}(W) \approx 10^{-9}$ and $h_{\mathrm{poly}}(W) \approx 10^{-14}$, i.e., cycles of length at least 13 would be numerically undetected by $h_{\mathrm{expm}}$ and $h_{\mathrm{poly}}$. In contrast, we observe that the value of the log-det function remains bounded away from zero and is able to detect larger cycles.

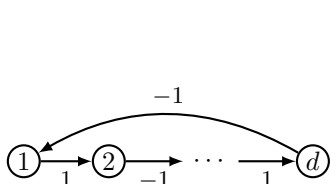 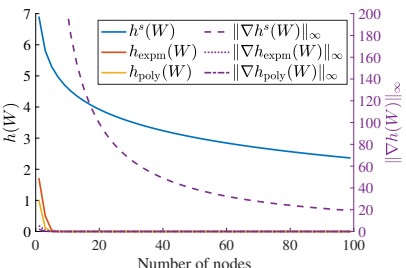

Figure 2: The values of $h_{\mathrm{expm}}$ and $h_{\mathrm{poly}}$ get very close to zero for a number of nodes as small as ten. In contrast, letting $s = 1.001$, we observe that $h_{\mathrm{ldet}}^s$ can stay away from zero even for a cycle graph of 100 nodes. Finally, we note a similar pattern for the entrywise $\ell_\infty$ norm of the gradients.

At first glance, it might seem difficult to directly compare the value of $h_{\mathrm{ldet}}^s$ to $h_{\mathrm{expm}}$ and $h_{\mathrm{poly}}$. We next show that when $s = 1$, $h_{\mathrm{ldet}}^{s=1}$ is an upper bound to $h_{\mathrm{expm}}$ and $h_{\mathrm{poly}}$.

**Lemma 4.** *For all $W \in \mathbb{W}^{s=1}$, we have $h_{\mathrm{poly}}(W) \leq h_{\mathrm{expm}}(W) \leq h_{\mathrm{ldet}}^{s=1}(W)$.*

The lemma above shows that in spite of $h_{\mathrm{ldet}}^{s=1}$, $h_{\mathrm{expm}}$, and $h_{\mathrm{poly}}$ being exact acyclicity characterizations, $h_{\mathrm{ldet}}^{s=1}$ will attain the largest value.

**Argument (ii).** $h_{\mathrm{ldet}}^s$ **has better behaved gradients.** Similar to argument (i), we show in Appendix B that $h_{\mathrm{expm}}$ and $h_{\mathrm{poly}}$ are susceptible to vanishing gradients even when the graph contains cycles (see Figure 2). The following lemma states that the magnitude of each entry of $\nabla h_{\mathrm{ldet}}^s$ at least as large as the magnitude of the corresponding entry of $\nabla h_{\mathrm{expm}}$ and $\nabla h_{\mathrm{poly}}$, and hence, $h_{\mathrm{ldet}}$ has larger gradients to guide optimization.

**Lemma 5.** *For any walk of length $k$, its contribution to the gradients $\nabla h_{\mathrm{expm}}(W)$ and $\nabla h_{\mathrm{poly}}(W)$ are diminished by $1/k!$ and $\binom{d-1}{k}/(d-1)^k$, respectively. In contrast, $\nabla h_{\mathrm{ldet}}^{s=1}(W)$ does not diminish any walk of any length. This implies that $|\nabla h_{\mathrm{poly}}(W)| \leq |\nabla h_{\mathrm{expm}}(W)| \leq |\nabla h_{\mathrm{ldet}}^{s=1}(W)|$.*

**Argument (iii). Computing $h_{\mathrm{ldet}}^s$ and $\nabla h_{\mathrm{ldet}}^s$ is empirically faster.** Even though $h_{\mathrm{ldet}}^s$, $h_{\mathrm{expm}}$, and $h_{\mathrm{poly}}$ all three share the same computational complexity of $\mathcal{O}(d^3)$, in practice $h_{\mathrm{ldet}}^s$ can be computed in about an order of magnitude faster than $h_{\mathrm{expm}}$ and $h_{\mathrm{poly}}$. In Figure 3, we compare the runtimes of $h_{\mathrm{ldet}}^s$, $h_{\mathrm{expm}}$ and $h_{\mathrm{poly}}$ for randomly generated matrices, where we observe that computing $h_{\mathrm{ldet}}^s$ can be 10x faster than $h_{\mathrm{expm}}$ and $h_{\mathrm{poly}}$. See Appendix B for further details.

## 4 Optimization

In the previous section we argued why the log-det function should be preferred in practice. Let $f_\theta$ denote a model with parameters $\theta$ for the functions $f_j$ in (1), e.g., neural networks as in [59]. In this section we turn to the problem of minimizing a given score function $Q(f_\theta; \boldsymbol{X})$ constrained to $h_{\mathrm{ldet}}^s(W(\theta)) = 0$. That is, we aim to solve:

$$\min_\theta Q(f_\theta; \boldsymbol{X}) + \beta_1 \|\theta\|_1 \quad \text{subject to } h_{\mathrm{ldet}}^s(W(\theta)) = 0, \tag{6}$$

where we include the $\ell_1$ regularizer to promote sparse solutions.

---

[3]Note that here the sign of an edge is not important since $W \circ W$ will have all edge weights equal to $+1$.

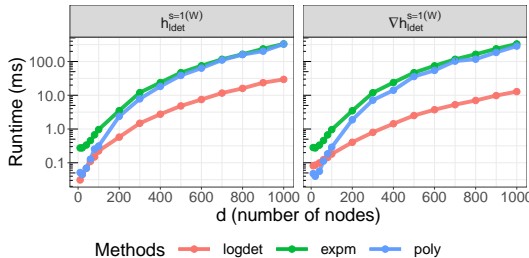

Figure 3: For each $d$, 30 matrices were randomly sampled from a standard Gaussian distribution.

Since the inception of the purely continuous framework for learning DAGs in [58], almost all follow-up work uses the augmented Lagrangian (AML) scheme to tackle problem (6), and L-BFGS-B [39] for solving the sequence of unconstrained problems. Motivated by the properties of $h_{\mathrm{ldet}}^s$ given in 3.1, we propose a simpler scheme named DAGMA, based on solving a sequence of unconstrained problems in which $h_{\mathrm{ldet}}^s$ is simply seen as a regularizer. DAGMA resembles the central path approach of barrier methods [8, 39], or the classical path-following approach for solving lasso problems [e.g. 14]. Our method is given in Algorithm 1.

---

**Algorithm 1** DAGMA

**Require:** Data matrix $\boldsymbol{X}$, initial central path coefficient $\mu^{(0)}$ (e.g., 1), decay factor $\alpha \in (0, 1)$ (e.g., 0.1), $\ell_1$ parameter $\beta_1 > 0$ (e.g., 0.01), log-det parameter $s > 0$ (e.g., 1), number of iterations $T$.
  1: Initialize $\theta^{(0)}$ so that $W(\theta^{(0)}) \in \mathbb{W}^s$.
  2: **for** $t = 0, 1, 2, \ldots T - 1$ **do**
  3:   Starting at $\theta^{(t)}$, solve $\theta^{(t+1)} = \arg\min_\theta \mu^{(t)}(Q(f_\theta; \boldsymbol{X}) + \beta_1 \|\theta\|_1) + h_{\mathrm{ldet}}^s(W(\theta))$
  4:   Set $\mu^{(t+1)} = \alpha\mu^{(t)}$
**Ensure:** $W(\theta^{(T)})$

---

The following lemma states that DAGMA will return a DAG at the limit of the central path. This is a critical distinction against existing methods, many of which rely on some type of post-processing (e.g. thresholding) to ensure that the solution is a DAG.

**Lemma 6.** *Algorithm 1 is guaranteed to return a DAG whenever $\mu^{(t)} \to 0$.*

### 4.1 Practical Considerations

  1. As in barrier methods, where it is required to start at the interior of the feasibility region, in Algorithm 1, we require that the initial point $W(\theta^{(0)})$ be inside $\mathbb{W}^s$. This is very easy to achieve since the zero matrix is in the interior of $\mathbb{W}^s$ for any $s > 0$; therefore, in our experiments we simply set $\theta^{(0)} = 0$.

  2. Note that in Algorithm 1, we let $\mu^{(t)}$ decrease by a constant factor at each iteration; however, it is possible to specify explicitly the value of each $\mu^{(t)}$, e.g., for $T = 4$, we can let $\mu = \{1, 0.1, 0.001, 0\}$.

  3. Regarding the choice of $s$, in principle $s$ could take any value greater than zero since DAGs are inside $\mathbb{W}^s$ for any $s > 0$. Similar to $\mu$, it is also possible to let $s$ vary at each iteration, e.g., for $T = 4$, we can set $s = \{1, 0.9, 0.8, 0.8\}$. In practive, we observe that *slightly* decreasing $s$ can help to obtain larger gradients as $W$ gets closer to a DAG. Note, however, that letting $s$ be equal or close to 1 is generally easier to optimize than setting $s$ closer to zero, the reason being that for smaller values of $s$ the volume of $\mathbb{W}^s$ is smaller and will require much smaller learning rates to stay inside $\mathbb{W}^s$, hence affecting convergence.

  4. Finally, we do not specify how to solve line 3 in Algorithm 1, this is because we leave the door open for different solvers to be used. For our experiments in the next section, we solve line 3 by using a first-order method with the ADAM optimizer [24], which works remarkably well as shown in our experiments. It remains as future work to exploit the Hessian structure of $h_{\mathrm{ldet}}$ given in Lemma 3 for second-order methods.

# 5 Experiments

We compare our method against GES [11], PC [51], NOTEARS [58], and GOLEM [37] on both *linear* and *nonlinear* SEMs. In Appendix C, we specify which existing implementation we used for each of the aforementioned methods. Consistent with previous work in this area (e.g., NOTEARS and follow up work), we have not performed any hyperparameter optimization: This is to avoid presenting unintentionally biased results. As a concrete example, for each of the SEM settings, we simply chose a reasonable value for the $\ell_1$ penalty coefficient and used that same value for all graphs across many different numbers of nodes.

Our experimental setting is similar to [58, 59]. For the main text, we present only a small fraction of all our experiments. Moreover, since the accuracy of certain methods were significantly lower than other methods, we report results only against the most competitive ones; full results for all settings and methods can be found in Appendix C.

**Linear Models.** In Appendix C.1, we report results for linear SEMs with Gaussian, Gumbel, and exponential noises, and use the least squares loss. For small to moderate number of nodes, see Appendix C.1.1; for large number of nodes, see Appendix C.1.2; for denser graphs, see Appendix C.1.4; and for a comparison against GOLEM for sparser graphs, see Appendix C.1.3.

**Nonlinear Models.** In Appendix C.2, we report results for nonlinear SEMs with binary and continuous data. For binary data, we use a logistic model for each structural equation, and use the log-likelihood loss as the score, we report results for small to large number of nodes in Appendix C.2.1. For continuous data, we consider the continuous additive noise model with Gaussian noise [9], where each nonlinear relationship is modeled by a multilayer perceptron, and use the log-likelihood loss as the score, we report results for small to moderate number of nodes in Appendix C.2.2.

In the following figures, ER4 and SF4 denote Erdős-Rényi and scale-free graphs, respectively, where for each number of nodes $d$, each graph has in expectation $4d$ edges. It is worth noting that the empirical settings by Zheng et al. [58, 59] consider graph models such as ER1, ER2, SF1, and SF2. Here we focus on the **hardest** setting, i.e., **ER4** and **SF4** graphs. For linear SEMs, Figure 4 shows results for graphs with $d \in [20, 100]$, and Figure 5 shows results for graphs with $d \in [200, 1000]$. In both regimes, we note that DAGMA obtains significant speedups and improvements in terms of structural accuracy (SHD) against NOTEARS and GOLEM, even though GOLEM is *specific* to and specialized for linear Gaussian SEMs. For nonlinear SEMs, Figure 6 shows results for logistic models, we similarly observe that DAGMA attains major speedups and improvements on SHD against NOTEARS [58]. Finally, for nonlinear models using neural networks, we observe that DAGMA is comparable in SHD to the NONLINEAR NOTEARS [59] but obtains significant speedups. Again, we invite the reader to look at Appendix C for more details and additional experiments.

# 6 Final Remarks

A relevant assumption in this work is that of sufficiency, that is, there are no hidden variables that are a common cause of at least two observed variables. While this assumption is widely used for structure learning, we nonetheless highlight that in practice it is very difficult to find scenarios where such assumption holds. As with all work that assumes sufficiency, our work is an important necessary step to understanding settings with hidden variables. Finally, we note that the work by [6] proposes a differentiable approach for ADMG for the semi-Markovian case using $h_{\mathrm{expm}}$. It is left for future work to explore the performance of such method using $h_{\mathrm{ldet}}$.

Another important limitation of this and previous work on the continuous framework for learning DAGs is that of providing guarantees on the learned structure. As in real-life applications one does not have access to the ground-truth DAG, there is much uncertainty as to whether an edge in the predicted DAG actually corresponds to a causal relation. Thus, there is still a need for formal guarantees under the continuous framework.

## Acknowledgments and Disclosure of Funding

K. B. was supported by NSF under Grant # 2127309 to the Computing Research Association for the CIFellows 2021 Project. B.A. was supported by NSF IIS-1956330, NIH R01GM140467, and the Robert H. Topel Faculty Research Fund at the University of Chicago Booth School of Business. P.R. was supported by ONR via N000141812861, and NSF via IIS-1909816, IIS-1955532, IIS-2211907.

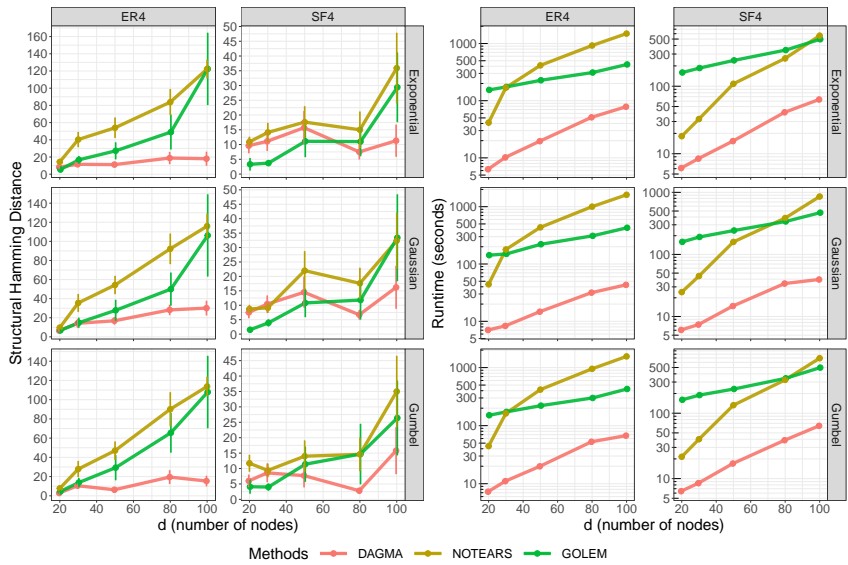

Figure 4: Experiments on linear SEMs for $d \in [20, 100]$. Each point in the plot is estimated over 10 repetitions, where error bars are the standard error. Wall time limit was set to 36 hours.

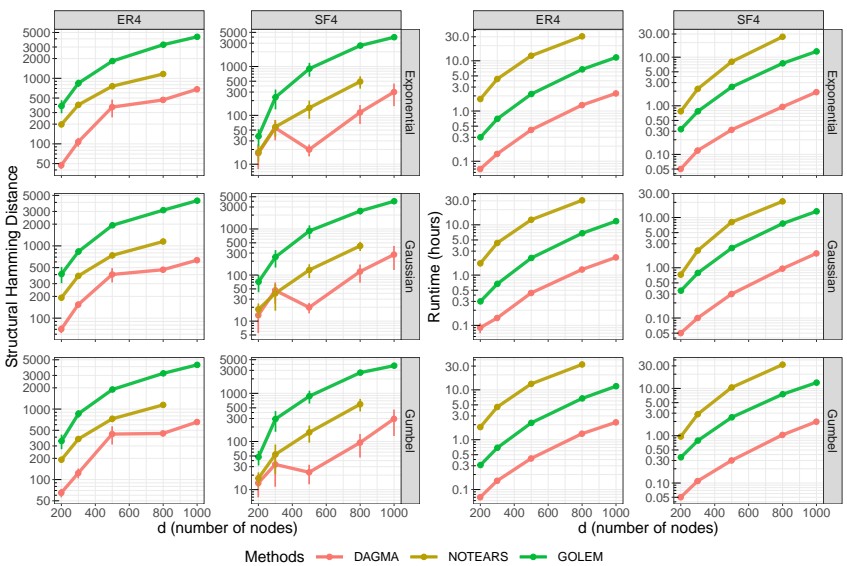

Figure 5: Experiments on linear SEMs for $d \in [200, 1000]$. Each point in the plot is estimated over 10 repetitions, where error bars are the standard error. Wall time limit was set to 36 hours.

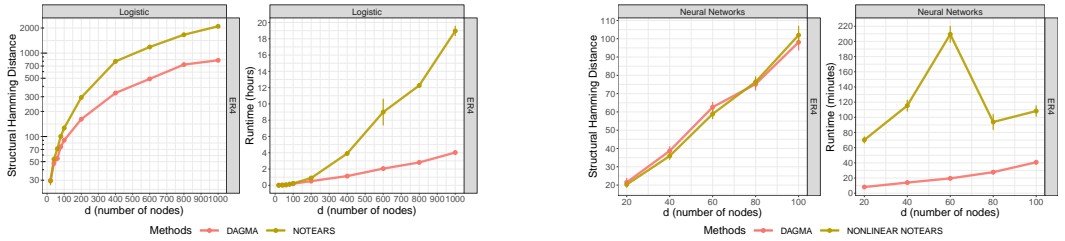

Figure 6: Experiments on nonlinear SEMs. Each point in the plot is estimated over 10 repetitions, where error bars are the standard error. Wall time limit was set to 36 hours.

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
