# SUPPLEMENTARY MATERIAL
# DAGMA: Learning DAGs via M-matrices and a Log-Determinant Acyclicity Characterization

## A  Detailed Proofs

### A.1  Proof of Theorem 1

*Proof.* We first note that for any $s > 0$ and matrix $A \in \mathbb{R}^{d \times d}$ we have $\det(sA) = s^d A$. Then, $\log \det(sI - W \circ W) - d \log s = \log(s^d \det(I - s^{-1}W \circ W)) - d \log s = \log \det(I - s^{-1}W \circ W)$. Moreover, since $W \in \mathbb{W}^s$, we have that $s > \rho(W \circ W)$ or equivalently $1 > \rho(s^{-1}W \circ W)$. Thus, in the sequel of the proof we set $s = 1$ w.l.o.g.

**Item (ii).** The gradient expression follows from standard matrix calculus [47]. From Lemma 2, it follows that $W$ is an stationary point of $h_{\text{ldet}}^{s=1}(W)$, i.e., $\nabla h_{\text{ldet}}^{s=1}(W) = 0$, if and only if $W$ corresponds to a DAG.

**Item (i).** From the item above, we characterized the stationary points of $h_{\text{ldet}}^{s=1}$. Moreover, from Proposition 1, we know that for any $W \in \mathbb{W}^s$, the gradient $\nabla h_{\text{ldet}}^{s=1}$ is well-defined since $I - W \circ W$ is an M-matrix and, thus, its inverse exists. Finally, note that at the boundary of $\mathbb{W}^s$, we have that $h_{\text{ldet}}^{s=1}(W) \to \infty$. From these observations, we have that the global minima of $h_{\text{ldet}}^{s=1}$ must be in the interior of $\mathbb{W}^s$ and will correspond to the set of stationary points. Hence, DAGs are local and global minima of $h_{\text{ldet}}^{s=1}$.

We conclude by noting that if $W$ is a DAG then we have $\det(I - W \circ W) = 1$ and the equality $h_{\text{ldet}}^{s=1}(W) = 0$ holds immediately. Since DAGs are global minima, this implies that for all $W \in \mathbb{W}^s$ we have $h_{\text{ldet}}^{s=1}(W) \geq 0$. □

### A.2  Proof of Lemma 1

*Proof.* **Item (i).** The proof follows directly from the fact that the weighted adjacency matrix $W$ of any DAG is a nilpotent matrix. Since $W \circ W$ is also a nilpotent matrix, its spectral radius is zero, i.e., $\rho(W \circ W) = 0$. Thus, for any $s > 0$, if $W$ is a DAG then $W \in \mathbb{W}^s$.

**Item (ii).** Recall that a space $\mathcal{X}$ is path-connected if, for any two points $x, y \in \mathcal{X}$, there exists a continuous function (path) $\phi : [0, 1] \to \mathcal{X}$ such that $\phi(0) = x$ and $\phi(1) = y$. Note that since the $d \times d$ zero matrix, $\mathbf{0}$, has spectral radius zero, it clearly follows that $\mathbf{0} \in \mathbb{W}^s$ for any $s > 0$. We prove path-connectedness of $\mathbb{W}^s$ by showing that for any $W \in \mathbb{W}^s$ there exist a path $\phi$ to $\mathbf{0}$. Then, for any $W \in \mathbb{W}^s$, define $\phi(t) = (1 - t)W$. It is clear that $\phi$ is a continuous function on $t$, where $\phi(0) = W \in \mathbb{W}^s$ and $\phi(1) = 0 \in \mathbb{W}^s$. Now we need to show that $\phi(t) \in \mathbb{W}^s$ for all $t \in (0, 1)$. Let $t_1$ be an arbitrary number in $(0, 1)$, by the nonnegativity of $W \circ W$ and by $(1 - t_1)^2 < 1$, we have that $(1 - t_1)^2 W \circ W < W \circ W$. Finally, by Perron-Frobenius theory on nonnegative matrices, we have that $\rho\left((1 - t_1)^2 W \circ W\right) < \rho(W \circ W) < s$, where the last inequality follows by $W \in \mathbb{W}^s$, which implies that $\phi(t_1) \in \mathbb{W}^s$. As the choice of $t_1$ was arbitrary, we conclude the proof.

**Item (iii).** The proof follows immediately by the definition of $\mathbb{W}^s$. □

### A.3  Proof of Lemma 2

*Proof.* First, recall that $\nabla h_{\text{ldet}}^s(W) = 2(sI - W \circ W)^{-\top} \circ W$. Second, since $(sI - W \circ W)$ is an M-matrix, by Proposition 1, we have that $(sI - W \circ W)^{-\top} \geq 0$. By the latter, whenever $[\nabla h_{\text{ldet}}^s(W)]_{i,j} \neq 0$, we have that $\text{sign}([\nabla h_{\text{ldet}}^s(W)]_{i,j}) = \text{sign}(W_{i,j})$. Finally, from the series expansion of the inverse we have:

$$(sI - W \circ W)^{-1} = \frac{1}{s}I + \frac{1}{s^2}(W \circ W) + \frac{1}{s^3}(W \circ W)^2 + \cdots,$$

by taking the transpose, that implies that the $i, j$ entry $[(sI - W \circ W)^{-\top}]_{i,j}$ is nonzero if and only if there exists a directed walk from $j$ to $i$. By taking the Hadamard product, we have that $[(sI - W \circ W)^{-\top} \circ W]_{i,j}$ is nonzero if and only if $W_{i,j} \neq 0$ and $[(sI - W \circ W)^{-\top}]_{i,j} \neq 0$, i.e., there must exist a closed walk of the form $i \rightarrow j \rightsquigarrow i$. Which concludes the proof. $\qquad\square$

### A.4 Proof of Lemma 3

*Proof.* We use the Magnus-Neudecker convention [28] for calculating the Hessian. Then, by taking differentials and vectorizing, we obtain:

$$\partial(\nabla h(W)) = 2\, \partial N^\top \circ W + 2\, N^\top \circ \partial W$$

$$\mathrm{vec}(\partial(\nabla h(W))) = 2\, \mathrm{vec}(\partial N^\top \circ W) + 2\, \mathrm{vec}(N^\top \circ \partial W)$$

$$= 2\, \mathrm{Diag}(\mathrm{vec}(W))\mathrm{vec}(\partial N^\top) + 2\, \mathrm{Diag}(\mathrm{vec}(N^\top))\mathrm{vec}(\partial W). \qquad (7)$$

Recall that $N = (sI - W \circ W)^{-1}$, we now derive the expression for $\mathrm{vec}(\partial N^\top)$,

$$\mathrm{vec}(\partial N^\top) = -(N \otimes N^\top)\, \mathrm{vec}(\partial(sI - W \circ W)^\top)$$

$$= 2\, (N \otimes N^\top)\, \mathrm{vec}(W^\top \circ \partial W^\top)$$

$$= 2\, (N \otimes N^\top)\, \mathrm{Diag}(\mathrm{vec}(W^\top))\mathrm{vec}(\partial W^\top)$$

$$= 2\, (N \otimes N^\top)\, \mathrm{Diag}(\mathrm{vec}(W^\top))K^{dd}\, \mathrm{vec}(\partial W),$$

plugging in the last equality into eq.(7), we have

$$\nabla^2 h(W) = \frac{\partial\, \mathrm{vec}(\nabla h(W))}{\partial\, \mathrm{vec}(W)}$$

$$= 4\, \mathrm{Diag}(\mathrm{vec}(W))(N \otimes N^\top)\, \mathrm{Diag}(\mathrm{vec}(W^\top))K^{dd} + 2\, \mathrm{Diag}(\mathrm{vec}(N^\top)),$$

which concludes the proof. $\qquad\square$

### A.5 Proof of Lemma 4

*Proof.* The comparison between $h_{\mathrm{expm}}$ and $h_{\mathrm{poly}}$ is straightforward by looking at the coefficients of their series expansions. Recall that, $h_{\mathrm{expm}}(W) = \sum_{k=0}^{\infty} 1/k!\, \mathrm{Tr}((W \circ W)^k) - d$ and $h_{\mathrm{poly}}(W) = \sum_{k=0}^{d} \binom{d}{k}/d^k\, \mathrm{Tr}((W \circ W)^k) - d$. Since $1/k! \geq \binom{d}{k}/d^k$, it is clear that $h_{\mathrm{expm}}(W) \geq h_{\mathrm{poly}}(W)$. To prove that $h_{\mathrm{ldet}}^{s=1}(W) \geq h_{\mathrm{expm}}(W)$, we use the fact that every square matrix has a Jordan canonical form. Let $W \circ W = Q^{-1}JQ$, where $Q$ is an invertible matrix and $J$ is in Jordan normal form (i.e., a block diagonal matrix with 1s in the super-diagonal), we have that $\mathrm{Tr}(e^{W \circ W}) = \mathrm{Tr}(e^J)$. Let $\Lambda(W \circ W)$ be the set of distinct generalized eigenvalues of $W \circ W$, and $m_\lambda$ be the multiplicity corresponding to $\lambda \in \Lambda(W \circ W)$. Then, we have that

$$h_{\mathrm{expm}}(W) = \mathrm{Tr}(e^{W \circ W}) - d = \sum_{\lambda \in \Lambda(W \circ W)} m_\lambda(e^\lambda - 1).$$

From $h_{\mathrm{ldet}}^{s=1}(W)$ we have,

$$h_{\mathrm{ldet}}^{s=1}(W) = -\log\det(I - W \circ W) = \sum_{\lambda \in \Lambda(W \circ W)} m_\lambda(-\log(1 - \lambda)),$$

where $\log$ denotes the principal branch of the complex logarithm. For any complex $\lambda$, we have the Taylor series:

$$e^\lambda - 1 = \sum_{k=1}^{\infty} \frac{\lambda^k}{k!}, \qquad -\log(1 - \lambda) = \sum_{k=1}^{\infty} \frac{\lambda^k}{k},$$

where both series converges precisely for all complex numbers $|\lambda| \leq 1, \lambda \neq 1$, which is the case as $W \in \mathbb{W}^{s=1}$. From the latter, it is clear to see that $h_{\mathrm{expm}}(W) \leq h_{\mathrm{ldet}}^{s=1}(W)$, which concludes the proof. $\qquad\square$

### A.6 Proof of Lemma 5

*Proof.* First let us write the gradients for the different acyclicity characterizations. Then, we have $\nabla h_{\mathrm{expm}}(W) = 2(e^{W \circ W})^\top \circ W$, $\nabla h_{\mathrm{poly}}(W) = 2((I + \frac{1}{d}W \circ W)^{d-1})^\top \circ W$, and $\nabla h_{\mathrm{ldet}}^{s=1}(W) = 2((I - W \circ W)^{-1})^\top \circ W$. When taking absolute values, it is clear that they differ due to the left-hand side of each Hadamard product. Thus, we need to look at the entries of: $|e^{W \circ W}|$, $|(I + \frac{1}{d}W \circ W)^{d-1}|$, and $|(I - W \circ W)^{-1}|$. From their series expansions we have:

$$e^{W \circ W} = \sum_{k=0}^{\infty} \frac{1}{k!}(W \circ W)^k,$$

$$(I + \frac{1}{d}W \circ W)^{d-1} = \sum_{k=0}^{d-1} \frac{\binom{d-1}{k}}{(d-1)^k}(W \circ W)^k,$$

$$(I - W \circ W)^{-1} = \sum_{k=0}^{\infty}(W \circ W)^k.$$

Since $W \circ W$ is nonnegative, each power $(W \circ W)^k$ is also nonnegative. Therefore, by noting that $\binom{d-1}{k}/(d-1)^k \leq 1/k! \leq 1$ for all $k$, the statement follows. $\qquad\square$

### A.7 Proof of Lemma 6

*Proof.* The proof follows by noting that at the limit of the central path ($\mu^{(t)} \to 0$) we solve the following problem:

$$\widehat{\theta} = \arg \min_\theta h_{\mathrm{ldet}}^s(W(\theta)).$$

Then, by the invexity property of $h_{\mathrm{ldet}}^s$ (see Corollary 3), it follows that the solution $W(\widehat{\theta})$ must be a DAG. $\qquad\square$

### A.8 Proof of Corollary 1

*Proof.* For any $W \in \mathbb{W}^s$, from Lemma 2, we know that the nonzeros of $\nabla h_{\mathrm{ldet}}^s(W)$ have the same sign as the corresponding entries in $W$. Then, let $Y = W - a\nabla h_{\mathrm{ldet}}^s(W)$ for a small value $a$ such that $|Y| \leq |W|$. It follows that $Y \circ Y \leq W \circ W$. Since $Y \circ Y$ and $W \circ W$ are nonnegative matrices, by Perron-Frobenius, we have that $\rho(Y \circ Y) \leq \rho(W \circ W)$. The latter implies that $Y \in \mathbb{W}^s$, thus, the negative gradient, $-\nabla h_{\mathrm{ldet}}^s$, must point towards the interior of $\mathbb{W}^s$. $\qquad\square$

### A.9 Proof of Corollary 2

*Proof.* Given the Hessian expression in Lemma 3, the expressions for its entries follow by simple algebraic manipulation. From the argument in the proof of Lemma 2 (see Apeendix A.3), we have that $N_{i,j} > 0$ if and only if there exist a directed walk from $i$ to $j$. By the latter, the signs and cycle interpretations follow. $\qquad\square$

### A.10 Proof of Corollary 3

*Proof.* Follows from Theorem 1. $\qquad\square$

## B Additional Discussions

### B.1 Additional Example for Section 3.1

In Figure 1, we provided an example of a two-node graph to visualize the properties of $h_{\mathrm{ldet}}$. In Figure 7, we present another example for a three-node graph with three edges (parameters). Specifically, the graph is parameterized by $W = \begin{bmatrix} 0 & w_1 & 0 \\ 0 & 0 & w_2 \\ 0 & w_3 & 0 \end{bmatrix}$. Here note that for $W$ to be a DAG at least one of $w_2$ or $w_3$ must be zero.

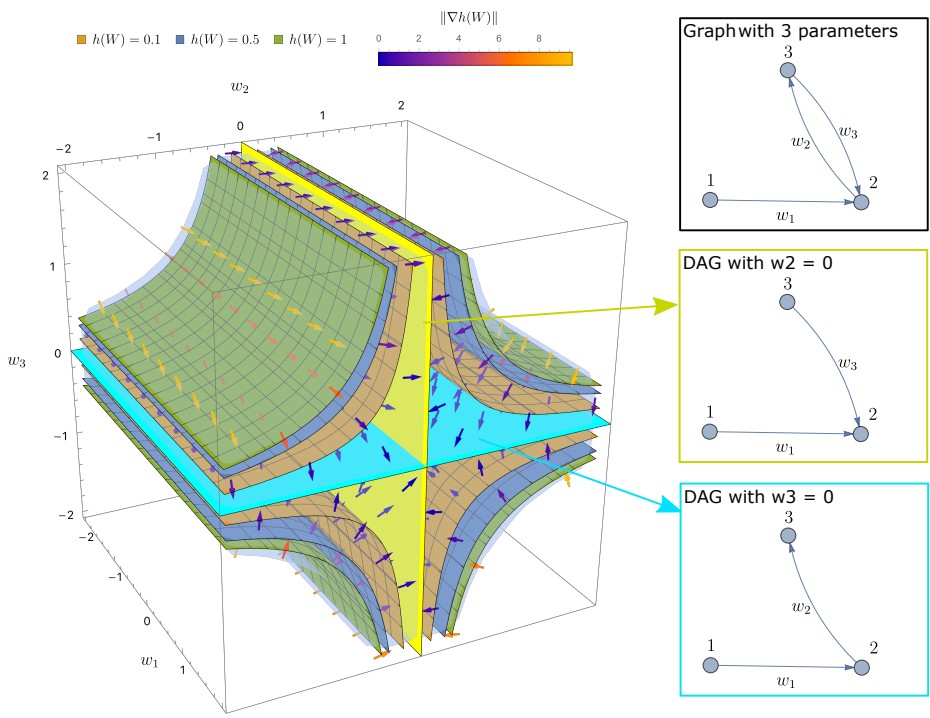

Figure 7: The curved manifolds represent the level sets of $h_{\mathrm{ldet}}^{s=1}$. The arrows represent the vector field $\nabla h_{\mathrm{ldet}}^{s=1}$. The yellow plane represents DAGs where $w_2 = 0$, while the cyan plane represents DAGs where $w_3 = 0$. Similar to Figure 1, we observe that the negative gradients point towards the interior of $\mathbb{W}^{s=1}$ and that DAGs are stationary points (attractors) of $h_{\mathrm{ldet}}^{s=1}$.

## B.2 Further Details for Section 3.2

In this section, we expand the discussion given in Section 3.2

**Argument (i).** $h_{\mathrm{ldet}}^s$ **does not diminish cycles of any length.** Let $h_{\mathrm{tinv}}(W) = \mathrm{Tr}((I - W \circ W)^{-1}) = \sum_{k=0}^{\infty} \mathrm{Tr}((W \circ W)^k) - d$. We note that $h_{\mathrm{tinv}}$ is another acyclicity characterization, previously considered by Zheng et al. [58]. From its series expansion, the reader might wonder if simply considering $h_{\mathrm{tinv}}(W)$ would be a better alternative than $h_{\mathrm{ldet}}(W)$, $h_{\mathrm{expm}}(W)$, and $h_{\mathrm{poly}}(W)$. The answer is *no*: While this alternative characterization does not suffer from the dimishing cycle problem, it is prone to numerical instability, which was already noted in [58]. This numerical instability is mainly due to the exploding gradients of $\mathrm{Tr}((I - W \circ W)^{-1})$, which we discuss in our next argument.

**Argument (ii).** $h_{\mathrm{ldet}}^s$ **has better behaved gradients.** Let us write the gradients of the different acyclicity characterizations. We have:

$$\nabla h_{\mathrm{expm}}(W) = 2(e^{W \circ W})^\top \circ W, \qquad \nabla h_{\mathrm{poly}}(W) = 2((I + \frac{1}{d}W \circ W)^{d-1})^\top \circ W,$$

$$\nabla h_{\mathrm{tinv}}(W) = 2((I - W \circ W)^{-2})^\top \circ W, \qquad \nabla h_{\mathrm{ldet}}^s(W) = 2((sI - W \circ W)^{-1})^\top \circ W.$$

From the series expansion for $(I - W \circ W)^{-2}$ we obtain:

$$(I - W \circ W)^{-2} = \sum_{k=0}^{\infty} (k+1)(W \circ W)^k.$$

Then, $\nabla h_{\mathrm{tinv}}$ is numerically unstable whenever $W \circ W$ has walks of moderate weight since all walks of length $k$ are now weighted by $k + 1$, thus, being prone to exploding gradients. Similar to argument (i), we have the following series expansions:

$$e^{W \circ W} = \sum_{k=0}^{\infty} \frac{1}{k!}(W \circ W)^k, \qquad (I + \frac{1}{d}W \circ W)^{d-1} = \sum_{k=0}^{d-1} \frac{\binom{d-1}{k}}{(d-1)^k}(W \circ W)^k,$$

and

$$(sI - W \circ W)^{-1} = \sum_{k=0}^{\infty} \frac{1}{s^{k+1}} (W \circ W)^k.$$

One can observe that for $s = 1$, the gradient of the log-determinant weights equally all cycles of any length, whereas $\nabla h_{\mathrm{expm}}$ and $\nabla h_{\mathrm{poly}}$ are again susceptible to the vanishing cycle problem and, thus, they might suffer from vanishing gradients. The plot in Figure 2 shows how the gradients of $h_{\mathrm{expm}}$ and $h_{\mathrm{poly}}$ decay at a very fast rate as the cycle graph has more nodes.

**Argument (iii). Computing $h_{\mathrm{ldet}}^s$ and $\nabla h_{\mathrm{ldet}}^s$ is empirically faster.** In Figure 3, we compared the runtimes of $h_{\mathrm{ldet}}^s$, $h_{\mathrm{expm}}$ and $h_{\mathrm{poly}}$ for randomly generated matrices. We used the benchmarking library from PyTorch for better runtime estimates over single threads. Experiments were conducted on an Intel Xeon processor E5 v4 with 2.40 GHz frequency.

The reason that computing $h_{\mathrm{ldet}}^s$ and $\nabla h_{\mathrm{ldet}}^s$ is faster is that it involves computing a log-determinant and a matrix inverse respectively, and both of these problems enjoy the large body of work on optimized libraries for matrix factorizations (e.g., LU decomposition) and linear-system solvers. In contrast, computing $h_{\mathrm{expm}}$ relies on a truncated Taylor series of the exponential whose degree is typically estimated using scaling [22, 1], and this requires several matrix-matrix multiplications. The matrix exponential is also a notoriously tricky object to compute [33]. Similar to $h_{\mathrm{expm}}$, computing $h_{\mathrm{poly}}$ also requires several matrix-matrix multiplications and, thus, both attain similar performances.

## C Detailed Experiments

**Computing.** All experiments were conducted on an 8-core Intel Xeon processor E5-2680v4 with 2.40 GHz frequency, and 32GB of memory. Each experiment had a wall time of 36 hours.

**Graph Models.** Each simulation in our experiments samples a graph from two well-known random graph models:

- **Erdos-Rényi** (ER) graphs: These are random graphs whose edges are added independently with equal probability. We use the notation ER$k$ to indicate that the graph model is an ER graph with $kd$ edges in expectation.

- **Scale-free** (SF) graphs: These are random graphs simulated according to the preferential attachment process [4]. We use the notation SF$k$ to indicate that the graph model is an SF graph with $kd$ edges in expectation and $\beta = 1$, where $\beta$ is the exponent used in the preferential attachment process. It is worth noting that since we consider directed graphs, this particular model corresponds to Price's model, a classical graph model for the growth of citation networks

Note that ER graphs are random *undirected* graphs. To produce a DAG, we draw a random permutation of numbers from 1 to $d$ and orient the edges respecting this vertex ordering. For the case of SF graphs, the edges are oriented each time a new node is attached, thus, the sampled graph is a DAG. Once the ground-truth DAG is generated, we need to simulate the structural equation model, where we provide experiments for linear and nonlinear SEMs.

**Remark 4.** *It is worth noting that the experimental settings by Zheng et al. [58, 59] consider graph models such as ER1, ER2, ER4, SF1, SF2, and SF4. Here we mainly focus in the **hardest** settings, that is, **ER4** and **SF4**.*

**Metrics.** We evaluate the performance of each algorithm with the following four metrics:

- **Structure Hamming distance (SHD)**: A standard measurement for structure learning that counts the total number of edges additions, deletions, and reversals needed to convert the estimated graph into the true graph.

- **True Positive Rate (TPR)**: Measures the proportion of *correctly* identified edges with respect to the total number of edges in the ground-truth DAG.

- **False Positive Rate (FPR)**: Measures the proportion of *incorrectly* identified edges with respect to the total number of *absent* edges in the ground-truth DAG.

- **Runtime**: Measures how much time the algorithm takes to run, we use it to measure the speed of the algorithms.

**Remark 5.** *Consistent with previous work in this area (e.g. NOTEARS and its follow-ups), we have not performed any hyperparameter optimization: This is to avoid presenting unintentionally biased results. As a concrete example, for each of the following SEM settings, we simply chose a reasonable value for the $\ell_1$ penalty coefficient and used that same value for all ER and SF graphs across many different numbers of nodes.*

## C.1   SEM: Linear Setting

In the linear case, the functions $f_j$ in (1) are directly parameterized by the weighted adjacency matrix $W$. That is, we have the following set of equations:

$$X_j = w_j^\top X + Z_j,$$

where $W = [w_1 \mid \cdots \mid w_d] \in \mathbb{R}^{d \times d}$, and $Z_j \in \mathbb{R}$ represents the noise. Here $W$ encodes the graphical structure, i.e., there is an edge $X_k \to X_j$ if and only if $W_{k,j} \neq 0$.

Then, given the ground-truth DAG $B \in \{0,1\}^{d \times d}$ from one of the two graph models ER or SF, we assigned edge weights independently from $\mathrm{Unif}\left([-2, -0.5] \cup [0.5, 2]\right)$ to obtain a weight matrix $W \in \mathbb{R}^{d \times d}$. Given $W$, we sampled $X = W^\top X + Z \in \mathbb{R}^d$ according to the following three noise models:

- **Gaussian noise:** $Z_j \sim \mathcal{N}(0,1), \forall j \in [d]$.
- **Exponential noise:** $Z_j \sim \mathrm{Exp}(1), \forall j \in [d]$.
- **Gumbel noise:** $Z_j \sim \mathrm{Gumbel}(0,1), \forall j \in [d]$.

Based on these models, we generated random datasets $\boldsymbol{X} \in \mathbb{R}^{n \times d}$ by generating the rows i.i.d. according to one of the models above. For each simulation, we generated $n = 1000$ samples, unless otherwise stated.

To measure the quality of a model, we use the least-square loss

$$Q(W; \mathbf{X}) = \frac{1}{2n} \|\mathbf{X} - \mathbf{X}W\|_F^2, \tag{8}$$

where $\|\cdot\|_F$ denotes the Frobenius norm. The coefficient $\beta_1$ used for $\ell_1$ regularization, and other parameters required for DAGMA (see Algorithm 1), are later specified for each figure.

The implementation details of the baselines are listed below:

- GES (specifically, the FGES algorithm in [48]) and PC [51] are standard baselines for structure learning. Their implementation is based on the `py-causal` package, available at https://github.com/bd2kccd/py-causal. The exact set of hyperparameters used are:
  - For PC: `testId = 'fisher-z-test', depth = 3, fasRule = 2, dataType = 'continuous', conflictRule = 1, concurrentFAS = True, useMaxPOrientationHeuristic = True`.
  - For GES: `scoreId = 'cg-bic-score', maxDegree = 5, dataType = 'continuous', faithfulnessAssumed = False`.
- The NOTEARS method in Zheng et al. [58] was implemented using the author's Python code available at: https://github.com/xunzheng/notears. Its score function is also the least square as defined in eq.(8). For the $\ell_1$ coefficient, for a fair comparison, we use the same value used for DAGMA. For the rest of hyperparameters, we use their default values.
- The GOLEM method in Ng et al. [37] was implemented using the author's Python and Tensorflow code available at: https://github.com/ignavierng/golem. Here we use their default set of hyperparameters, that is, $\lambda_1 = 0.02$ and $\lambda_2 = 5$, for other details of their method we refer the reader to Appendix F of Ng et al. [37].

### C.1.1 Small to Moderate Number of Nodes

Following the aforementioned process to generate data, in this section, we test the methods for graphs with number of variables $d \in \{20, 30, 50, 80, 100\}$. We use the following setting for DAGMA (Algorithm 1): Number of iterations $T = 4$, initial central path coefficient $\mu^{(0)} = 1$, decay factor $\alpha = 0.1$, $\ell_1$ coefficient $\beta_1 = 0.05$, log-det parameter $s = \{1, .9, .8, .7\}$. For each problem in line 3 of Algorithm 1, we implement an adaptive gradient method using the ADAM optimizer [24]. The hyperparameters for ADAM are: Learning rate of $3 \times 10^{-4}$, and $(\beta_1, \beta_2) = (0.99, 0.999)$. For $t = \{0, 1, 2\}$, we run ADAM for $2 \times 10^4$ iterations or until the loss converges, whichever comes first. For $t = 3$, we run ADAM for $7 \times 10^4$ iterations or until the loss converges, whichever comes first. We consider that the loss converges if the relative error between subsequent iterations is less than $10^{-6}$. Finally, as in previous work including the baseline methods [58, 59, 37], a final thresholding step is performed as it was shown to help reduce the number of false discoveries. For all cases, we use a threshold of 0.3.

The results for different graph models (ER4, SF4), and different noise distributions (Gaussian, Gumbel, exponential), are shown in Figure 8. In Table 1, we average the SHDs and runtimes across graph and noise types, for the competitive methods. Here we note in particular that for $d = 100$, DAGMA obtains an **improvement** of 74.9% and 76.5% in **SHD** against GOLEM and NOTEARS, respectively; also, DAGMA runs 7.7 and 19.1 times **faster** than GOLEM and NOTEARS, respectively. Finally, we note that DAGMA performs much better than GOLEM besides the latter being *tailored* to linear Gaussian models.

Table 1: Summary of performances (SHD and runtime) of the most competitive methods. Each metric was averaged across different graph and noise types. Finally, the errors denote 95% confidence intervals on 10 repetitions.

| Method | $d$ | SHD | Runtime (seconds) |
|---|---|---|---|
| **DAGMA** | 20 | 6.78±1.64 | 6.54± 0.42 |
| | 30 | 11.05±2.50 | 8.99± 0.62 |
| | 50 | 12.03±3.42 | 16.88±0.98 |
| | 80 | 13.92±4.44 | 41.55±3.30 |
| | 100 | 17.80±5.72 | 59.36±4.80 |
| **GOLEM** | 20 | 4.28±1.38 | 154.64±2.50 |
| | 30 | 9.48±3.10 | 177.21±5.62 |
| | 50 | 19.60±7.30 | 231.53±3.58 |
| | 80 | 33.68±12.87 | 324.41±7.10 |
| | 100 | 70.95±26.11 | 458.94±8.62 |
| **NOTEARS** | 20 | 10.53±1.58 | 32.41±3.72 |
| | 30 | 22.70±6.04 | 104.76±20.43 |
| | 50 | 34.82±7.96 | 278.13±40.14 |
| | 80 | 52.22±13.49 | 640.95±89.24 |
| | 100 | 75.87±13.97 | 1129.10±120.74 |

### C.1.2 Large Number of Nodes

In this section, we test DAGMA, GOLEM, and NOTEARS for graphs with number of variables $d \in \{200, 300, 500, 800, 1000\}$. We do not test PC and GES as they are not competitive in terms of accuracy, as shown in Figure 8. We follow the same setting for DAGMA given in Section C.1.1.

The results for different graph models (ER4, SF4), and different noise distributions (Gaussian, Gumbel, exponential), are shown in Figure 9. In Table 2, we average the SHDs and runtimes across graph and noise types. Here we note in particular that for $d = 800$, DAGMA obtains an **improvement** of 90.2% and 65.5% in **SHD** against GOLEM and NOTEARS, respectively; also, DAGMA runs 6.2 and 25 times **faster** than GOLEM and NOTEARS, respectively. For $d = 1000$, we observe that NOTEARS takes more than 36 hours, which is the reason we could not report its performance. Finally, we note that once again DAGMA performs much better than GOLEM besides the latter being *tailored* to linear Gaussian models.

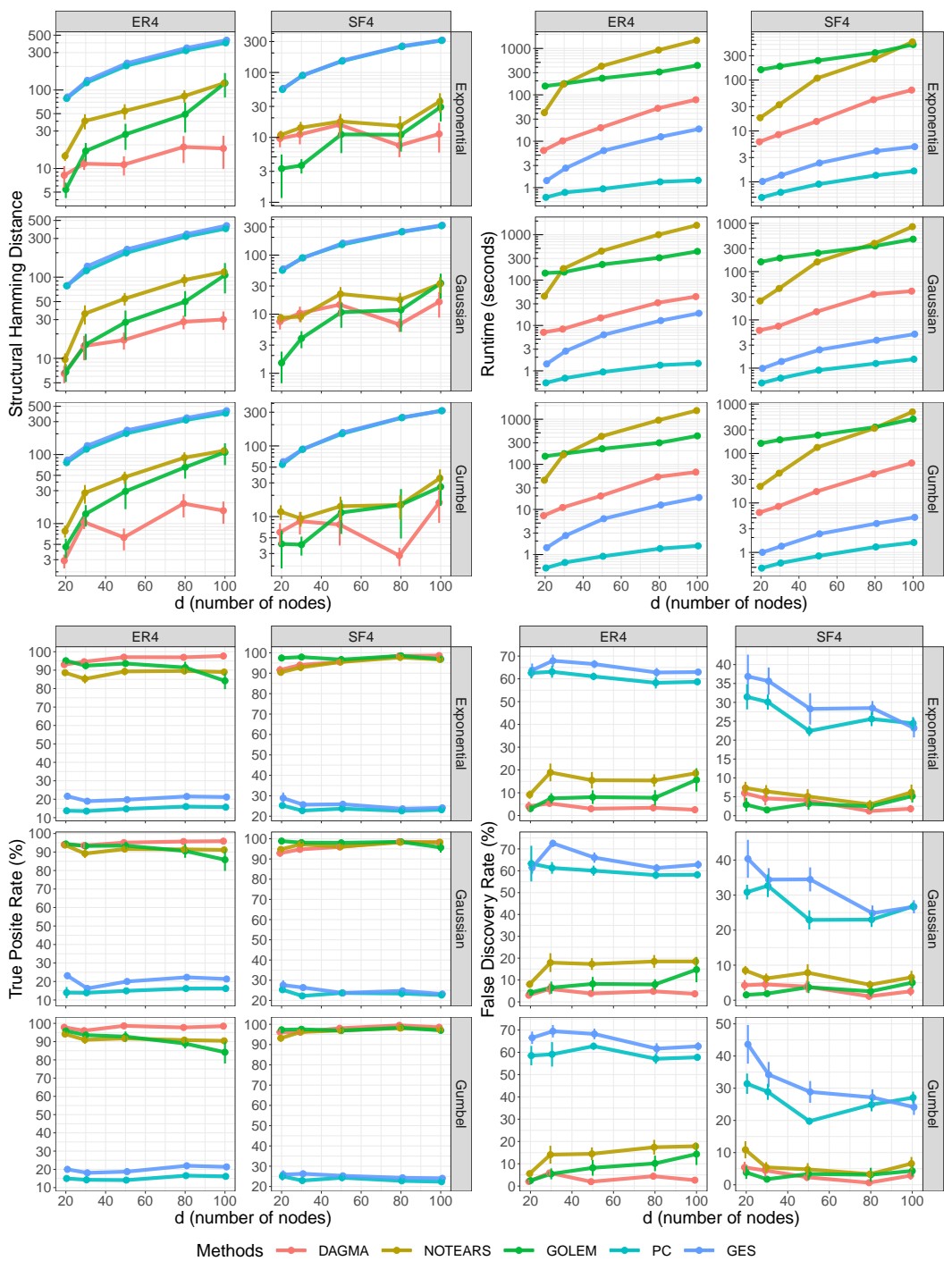

Figure 8: SHD, Runtime, TPR and FDR of all methods for different graph types (ER4, SF4) and different noise types (Gaussian, Gumbel, exponential). In all cases, lower is better *except* for the TPR. Error bars represent standard errors over 10 simulations. More details are given in Section C.1.1, and a summary is provided in Table 1.

### C.1.3 DAGMA vs GOLEM in Sparser Graphs

We note that in the work by Ng et al. [37], the authors performed experiments on large number of nodes **only for ER2 graphs**, that is, sparser graphs. It was reported in Ng et al. [37] that their

Table 2: Summary of performances (SHD and runtime) of the most competitive methods. Each metric was averaged across different graph and noise types. Finally, the errors denote $95\%$ confidence intervals on 10 repetitions.

| Method | $d$ | SHD | Runtime (hours) |
|---|---|---|---|
| **DAGMA** | 200 | 37.77$\pm$8.80 | 0.06$\pm$0.00 |
| | 300 | 86.65$\pm$19.63 | 0.13$\pm$0.00 |
| | 500 | 211.68$\pm$78.88 | 0.37$\pm$0.02 |
| | 800 | 285.90$\pm$56.89 | 1.15$\pm$0.06 |
| | 1000 | 473.70$\pm$101.63 | 2.09$\pm$0.08 |
| **GOLEM** | 200 | 215.68$\pm$66.91 | 0.32$\pm$0.00 |
| | 300 | 552.72$\pm$113.60 | 0.73$\pm$0.02 |
| | 500 | 1390.43$\pm$213.77 | 2.32$\pm$0.04 |
| | 800 | 2919.14$\pm$191.38 | 7.13$\pm$0.10 |
| | 1000 | 4083.03$\pm$157.58 | 12.50$\pm$0.22 |
| **NOTEARS** | 200 | 105.58$\pm$24.19 | 1.28$\pm$0.14 |
| | 300 | 217.73$\pm$48.30 | 3.42$\pm$0.28 |
| | 500 | 441.72$\pm$93.01 | 10.84$\pm$0.62 |
| | 800 | 829.08$\pm$118.10 | 28.79$\pm$1.22 |
| | 1000 | $-$ | $> 36$ |

GOLEM method performed reasonably well for large number of nodes. However, as shown in Figure 9, for denser graphs such as ER4 and SF4, the performance of GOLEM degrades very fast as $d$ increases. In this section, we experiment with the same graph model as in [37], i.e., ER2, for $d \in \{200, 300, 500, 800, 1000, 2000\}$ We note that even though GOLEM is competitive in this regime, DAGMA still obtains *significant* improvements.

DAGMA was run under the same setting described in Section C.1.1. The results are reported in Figure 10. Here we note in particular that for $d = 1000$, DAGMA obtains an **improvement** of $22.5\%$ in **SHD**, and runs $8.5$ times **faster** than GOLEM, even though the latter is customized for linear models. For $d = 2000$, GOLEM took more than 36 hours per simulation, hence, we could not report its performance.

### C.1.4 DAGMA vs NOTEARS and GOLEM in Denser Graphs

For completeness, we run experiments on a denser graph type such as ER6 with Gaussian noise, for $d \in \{20, 40, 60, 80, 100\}$. DAGMA was run under the same setting described in Section C.1.1. The results are reported in Figure 11. Here we note in particular that for $d = 100$, DAGMA obtains an **improvement** of $73.1\%$ and $44.5\%$ in **SHD** against GOLEM and NOTEARS, respectively; also, DAGMA runs $4.8$ and $20.6$ times **faster** than GOLEM and NOTEARS, respectively. Finally, we note that even in the regime of small number of variables, GOLEM's performance degrades very fast for denser graphs, while DAGMA's performance remains the best among the three methods.

## C.2 SEM: Non-Linear Setting

### C.2.1 Logistic Model

Given the ground-truth DAG $B \in \{0,1\}^{d \times d}$ from an ER4 graph, we assigned edge weights independently from $\text{Unif}\left([-2, -0.5] \cup [0.5, 2]\right)$ to obtain a weight matrix $W \in \mathbb{R}^{d \times d}$. Given $W$, we sampled $X_j = \text{Bernoulli}(\exp(w_j^\top X)/(1 + \exp(w_j^\top X))), \forall j \in [d]$. Based on this model, we generated random datasets $\boldsymbol{X} \in \mathbb{R}^{n \times d}$ by generating the i.i.d. rows. For each simulation, we generated $n = 5000$ samples for graphs with $d \in \{20, 40, 60, 80, 100, 200, 400, 600, 800, 1000\}$ nodes.

To measure the quality of a model, we use the log-likelihood loss

$$Q(f, \mathbf{X}) = \frac{1}{n} \sum_{i=1}^{d} \mathbf{1}_n^\top \left(\log(\mathbf{1}_n + \exp(f_i(\mathbf{X}))) - \mathbf{x}_i \circ f_i(\mathbf{X})\right). \tag{9}$$

The implementation details of the baselines are listed below:

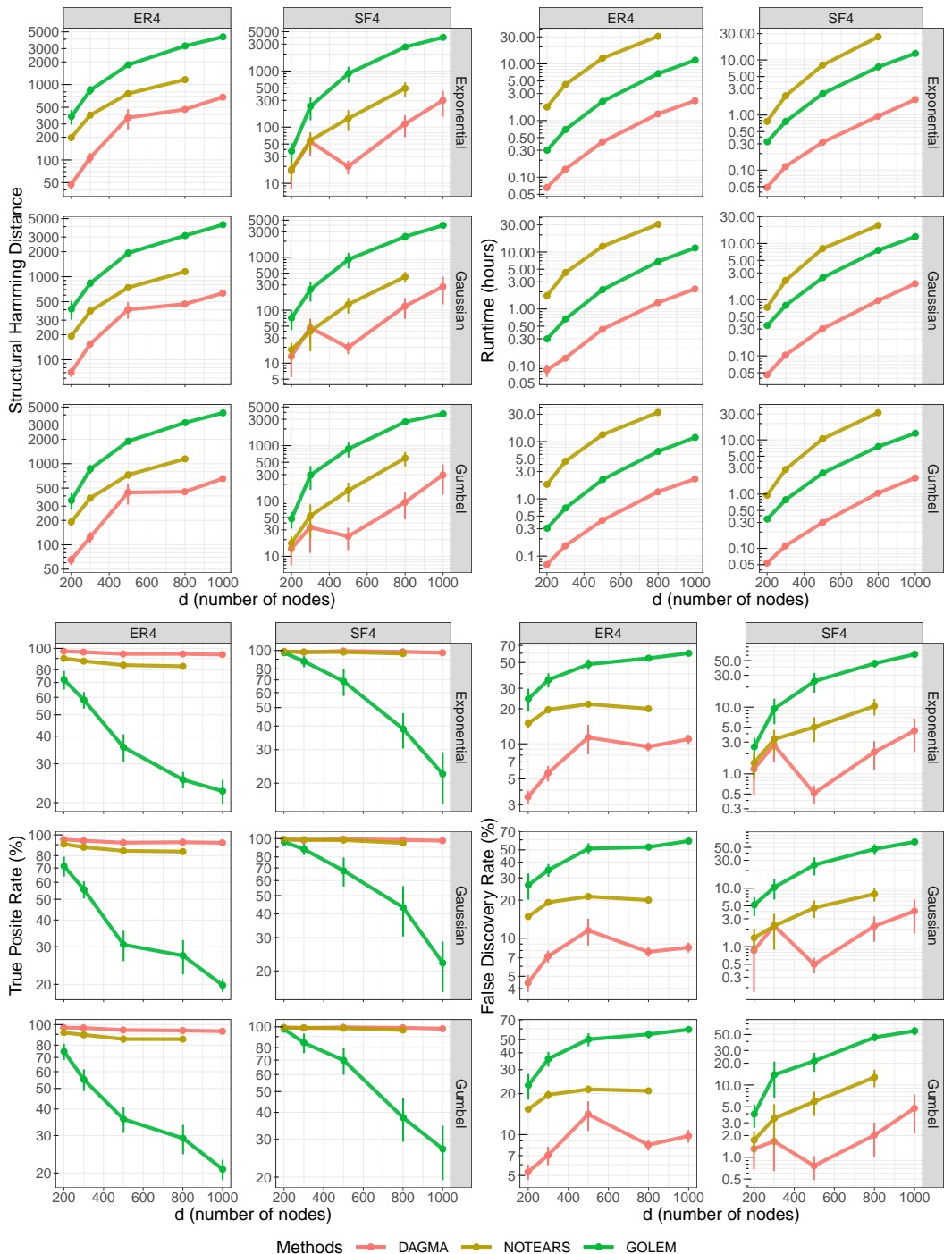

Figure 9: SHD, Runtime, TPR and FDR of competitive methods for different graph types (ER4, SF4) and different noise types (Gaussian, Gumbel, exponential). In all cases, lower is better *except* for the TPR. Error bars represent standard errors over 10 simulations. More details are given in Section C.1.2, and a summary is provided in Table 2.

- GES (specifically, the FGES algorithm in [48]) and PC [51] are standard baselines for structure learning. Their implementation is based on the py-causal package, available at https://github.com/bd2kccd/py-causal. The exact set of hyperparameters used are:

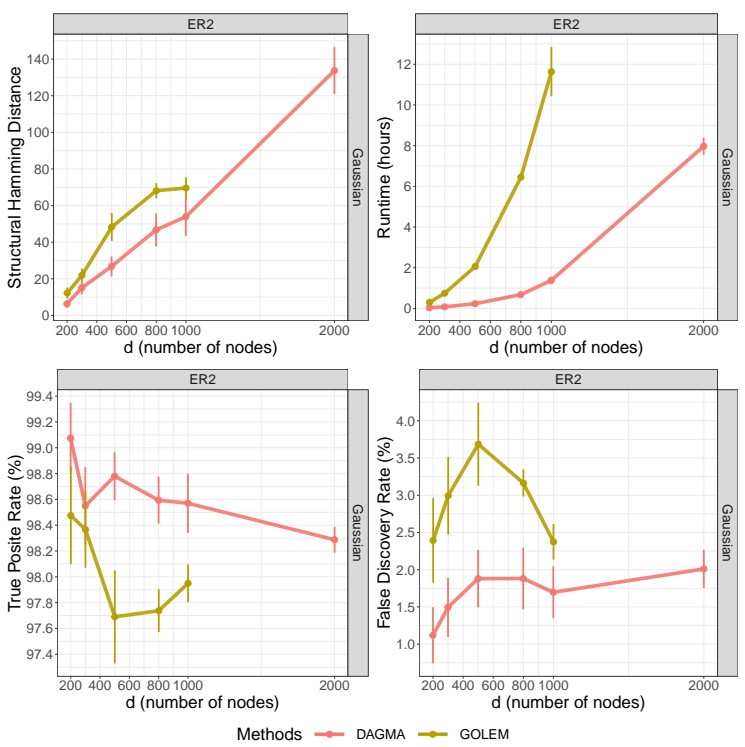

Figure 10: SHD, Runtime, TPR and FDR of DAGMA and GOLEM for a graph type ER2 with Gaussian noise. In all cases, lower is better *except* for the TPR. Error bars represent standard errors over 10 simulations. More details are given in Section C.1.3

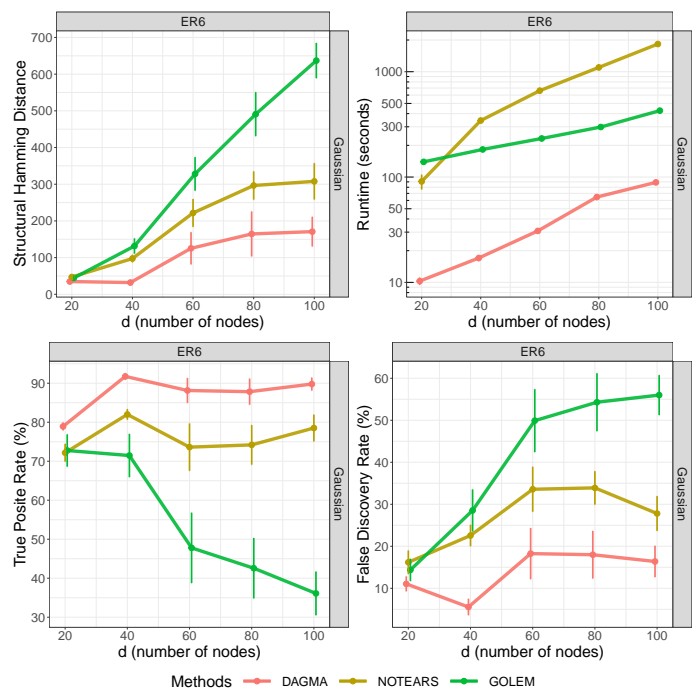

Figure 11: SHD, Runtime, TPR and FDR of DAGMA, GOLEM, and NOTEARS for a graph type ER6 with Gaussian noise. In all cases, lower is better *except* for the TPR. Error bars represent standard errors over 10 simulations. More details are given in Section C.1.4

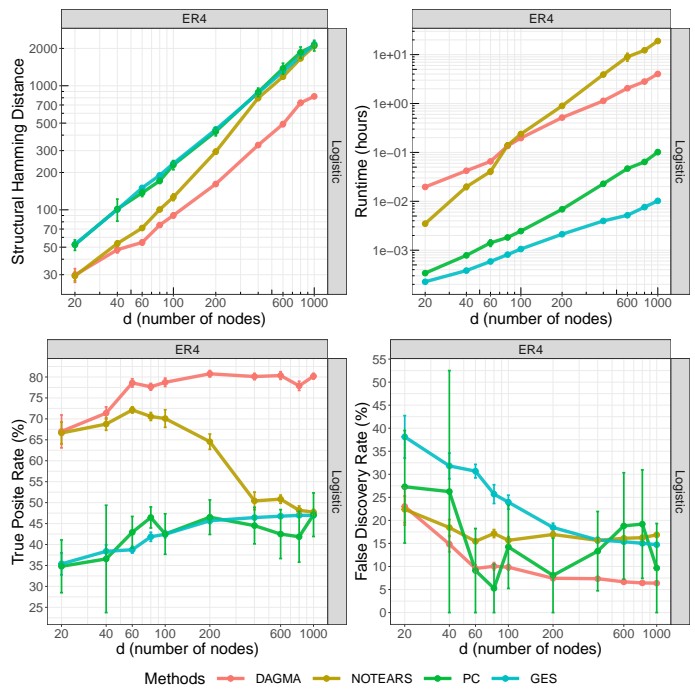

Figure 12: SHD, Runtime, TPR and FDR of all methods for a graph type ER4 and logistic model. In all cases, lower is better *except* for the TPR. Error bars represent standard errors over 10 simulations. More details are given in Section C.2.1

- For PC: `testId = 'disc-bic-test'`, `depth = 4`, `fasRule = 2`, `dataType = 'discrete'`, `conflictRule = 1`, `concurrentFAS = True`, `useMaxPOrientationHeuristic = True`.

- For GES: `scoreId = 'bdeu-score'`, `maxDegree = 5`, `dataType = 'discrete'`, `faithfulnessAssumed = False`.

- The NOTEARS method in Zheng et al. [58] was implemented using the author's Python code available at: https://github.com/xunzheng/notears. Its score function is also the log-likelihood loss as defined in eq.(9). For the $\ell_1$ coefficient, for a fair comparison, we use the same value used for DAGMA. For the rest of hyperparameters, we use their default values.

We use the following setting for DAGMA (Algorithm 1): Number of iterations $T = 4$, initial central path coefficient $\mu^{(0)} = 10$, decay factor $\alpha = 0.1$, $\ell_1$ coefficient $\beta_1 = 0.01$, log-det parameter $s = \{1, .9, .8, .7\}$. For each problem in line 3 of Algorithm 1, we implement an adaptive gradient method using the ADAM optimizer [24]. The hyperparameters for ADAM are: Learning rate of $3 \times 10^{-4}$, and $(\beta_1, \beta_2) = (0.99, 0.999)$. For $t = \{0, 1, 2\}$, we run ADAM for $10^4$ iterations or until the loss converges, whichever comes first. For $t = 3$, we run ADAM for $5 \times 10^4$ iterations or until the loss converges, whichever comes first. We consider that the loss converges if the relative error between subsequent iterations is less than $10^{-6}$. Finally, as in [58, 59, 37], a final thresholding step is performed as it was shown to help reduce the number of false discoveries. For all cases, we use a threshold of 0.3.

The results are shown in Figure 12. We note that for $d = 1000$, DAGMA obtains an **improvement** of 60% in **SHD** and runs 4.8 times **faster** than NOTEARS. Finally, we note that GOLEM is not considered for the nonlinear models as it only works for linear ones.

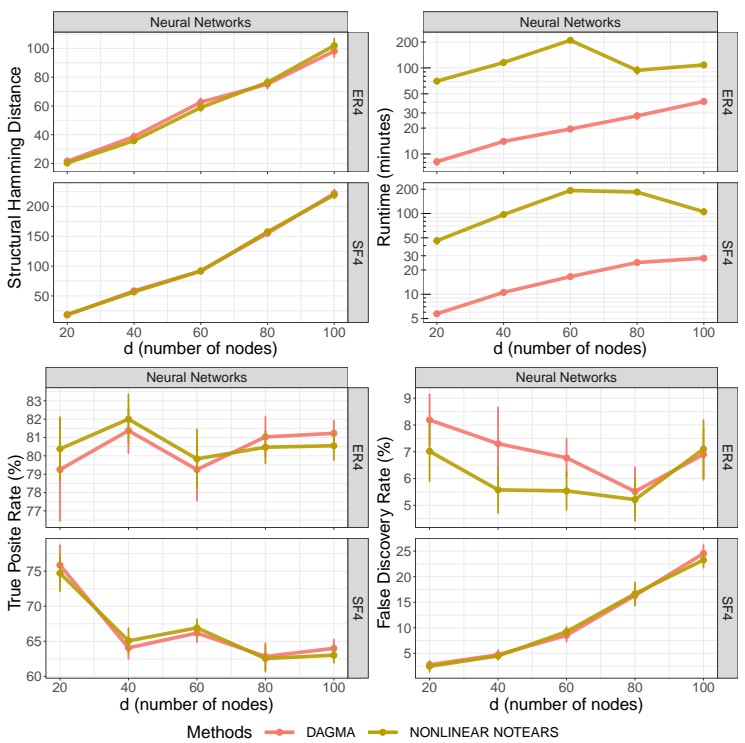

Figure 13: SHD, Runtime, TPR and FDR of all methods for a graph type ER4 and logistic model. In all cases, lower is better *except* for the TPR. Error bars represent standard errors over 10 simulations. More details are given in Section C.2.2

### C.2.2 Neural Network Model

We mainly follow the nonlinear setting of Zheng et al. [59]. That is, given a ground-truth graph $G$, we simulate the SEM:

$$X_j = f_j(X_{\text{pa}(j)}) + Z_j, \forall j \in [d],$$

where $Z_j \sim \mathcal{N}(0, 1)$ is a standard Gaussian noise. Here $f_j$ is a randomly initialized multilayer perceptron (MLP) with one hidden layer of size 100 and sigmoid activation. Similar to previous experiments, we generate a dataset $\boldsymbol{X} \in \mathbb{R}^{n \times d}$, with $n = 1000$ i.i.d. samples.

For this setting, we only compare to NONLINEAR NOTEARS [59]. We refer the reader to [59] for a comprehensive comparison with other baselines. For NONLINEAR NOTEARS and DAGMA, each $f_\theta$ is modeled by a MLP with one hidden layer of size 10 and sigmoid activation. In contrast to the original implementation of NONLINEAR NOTEARS [59] which uses the square loss, we use the log-likelihood as in [9] as we observe better performances for both methods.

We use the following setting for DAGMA (Algorithm 1): Number of iterations $T = 4$, initial central path coefficient $\mu^{(0)} = 0.1$, decay factor $\alpha = 0.1$, $\ell_1$ coefficient $\beta_1 = 0.02$, log-det parameter $s = 1$. For each problem in line 3 of Algorithm 1, we implement an adaptive gradient method using the ADAM optimizer [24]. The hyperparameters for ADAM are: Learning rate of $2 \times 10^{-4}$, and $(\beta_1, \beta_2) = (0.99, 0.999)$. For $t = \{0, 1, 2\}$, we run ADAM for $7 \times 10^4$ iterations or until the loss converges, whichever comes first. For $t = 3$, we run ADAM for $8 \times 10^4$ iterations or until the loss converges, whichever comes first. We consider that the loss converges if the relative error between subsequent iterations is less than $10^{-6}$. Finally, as in [58, 59, 37], a final thresholding step is performed as it was shown to help reduce the number of false discoveries. For all cases, we use a threshold of $0.3$.

The results are shown in Figure 13. We note that DAGMA and NOTEARS obtain similar performances in SHD; however, DAGMA can obtain 3x to 10x speedups over NONLINEAR NOTEARS. Finally, we note that GOLEM is not considered for the nonlinear models as it only works for linear ones.

# D  Broader Impacts

A potential misuse of this type of work would be to purposely (or not) run the method proposed on a dataset that is biased. Since we do not formally deal with inherent biases in the dataset (e.g., unfairness due to selection bias), it is possible to learn relationships that are not present in reality. A user can then (un)intentionally report a result incorrectly claiming to have found the cause of a certain variable, thus, creating misinformation.