# OpenReview forum: "DAGMA: Learning DAGs via M-matrices and a Log-Determinant Acyclicity Characterization"
_NeurIPS.cc/2022/Conference — NeurIPS 2022 Accept_

### Official Review · Reviewer_wwF9 · 2022-07-10

**Rating:** 8
**Confidence:** 3
**Soundness:** 4 excellent
**Presentation:** 3 good
**Contribution:** 4 excellent

**Summary:**

This paper proposes a new formulation promoting acyclicity. Going beyond the widely used approaches based on the trace of polynomials of the hadamard square of the adjacency matrix, this paper rather takes a different angle by considering a reduced set of matrices, M-matrices. This M-matrix based acyclicity regularizer is thoroughly investigated so that its empirically superior behavior is also understood theoretically.

**Questions:**

* In Corollary 1, what does it mean by 'point towards the interior'. Is this term 'interior' is the topological term or a general term for somewhere inside? It seems that $\mathbb{W}^s$ is an open subset of $R^{d \times d}$. Does this mean that its direction is always directly toward a stationary point?

* It seems that in eq(5) the condition that $W \ge 0$, that is, all entries are nonnegative is missing. It seems that some Theorem may hold without this nonnegative entry condition but others do not. Theorems/Lemma critically relying on Proposition 1 should require the nonnegative entry condition. However, for example, Lemma 2 seems to consider the cases without nonnegative entry conditions, otherwise, $sign(W_{i,j})$ is always nonnegative. It would be better if the authors clarify this.



**Limitations:**

* Except for general limitations of differentiable approaches for DAG discovery, there is no other specific limitation. To discuss such such general limitations some other references can be added. In Differentiable Causal Discovery Under Unmeasured Confounding; Rohit Bhattacharya, Tushar Nagarajan, Daniel Malinsky, Ilya Shpitser; AISTATS 2021. there are many interesting discussions and some solutions on such general limitations.
  * Differentiable approach for ADMG for semi-Markovian case
  * How to find MEC not a single DAG

**Strengths And Weaknesses:**

Strengths
* A novel approach based on M-matrix is proposed for an acyclicity regularizer free from optimization hassle in exiting approaches.
* The behavior of the log-det acyclicity regularizer -- the gradient and other regularization properties -- is analyzed to a quite detailed extent with supporting simulation results.
* DAG discovery with the proposed method outperforms others even with the advantage in the run-time.

Weaknesses
* For some theoretical results, it would be better if some technical detail is provided at a level to help to understand.
  * In line 177, $(4) \Rightarrow (3)$ required a bit of thought to understand so some brief comments would help instantaneous understanding.
  * In Lemma 3, the statement that $\otimes$ is the Kronecker product may be added so that reader can look up its definition.
* It seems that sometimes theorems assumes slightly different definition for $\mathbb{W}^2$ with/without nonnegative entry condition (details below)

---

> ### Author Response · Authors · 2022-08-02
> **Rebuttal**
>
> We thank the reviewer for appreciating the *novelty and relevance of our contributions* and for their positive evaluation of our work. We next address your comments in order.
>
> ### Strengths And Weaknesses:
>
> > ``In line 177, (4)⇒(3) required a bit of thought to understand so some brief comments would help instantaneous understanding.’’
>
> In Line 177, we simply present the domain $\mathbb{W}^s$ where (4) implies (3), but the formal statement is given in Theorem 1 item (i), which is proved in Section A.1. To make sure this is crystal clear, we will add a sentence stating that the implication is proved in Theorem 1.
>
> > In Lemma 3, the statement that ⊗ is the Kronecker product may be added so that reader can look up its definition.
>
> Agreed! We, unfortunately, missed specifying that ⊗ denotes the Kronecker product.
>
>
> ### Questions
>
> > In Corollary 1, what does it mean by 'point towards the interior'. Is this term 'interior' is the topological term or a general term for somewhere inside? It seems that W^s is an open subset of Rd×d.. Does this mean that its direction is always directly toward a stationary point?
>
> Yes, the term ‘interior’ is in the topological sense. It is true that since $\mathbb{W}^s$ is an open set, any element of $\mathbb{W}^s$ is an interior point of $\mathbb{W}^s$. While this is somewhat redundant, our aim was to emphasize that the direction of the negative gradient of $h^s_{\mathrm{ldet}}$ will provide a direction to remain inside $\mathbb{W}^s$. Note that we missed the word *"negative"* in Corollary 1, we will add this in the revision. Finally, the direction does not directly point towards a stationary point; however, a simple gradient descent scheme will lead to a stationary point of $h^s_{\mathrm{ldet}}$ which corresponds to a DAG, as prescribed by Theorem 1 and Corollary 2.
>
> > It seems that in eq(5) the condition that W≥0, that is, all entries are nonnegative is missing. It seems that some Theorem may hold without this nonnegative entry condition but others do not. Theorems/Lemma critically relying on Proposition 1 should require the nonnegative entry condition. However, for example, Lemma 2 seems to consider the cases without nonnegative entry conditions, otherwise, sign(Wi,j) is always nonnegative. It would be better if the authors clarify this.
>
> It is correct that many of our derivations rely on the properties of M-matrices given in Proposition 1. This is the reason our logdet function in Theorem 1 leverages the nonnegativity of the Hadamard product $W \circ W$. Then, we just need to make sure that the spectral radius of $W\circ W$ is less than $s$, which is precisely what eq.(5) is about! This should hopefully clarify your confusion about Lemma 2, if not, we are happy to address any other follow-up questions.
>
>
> ### Limitations
> Thank you for this suggestion! In fact, since Bhattacharya et al. also leverage differentiable acyclicity functions, it would be interesting to see the performance of our logdet formulation under the ADMG setting. We will add the reference for completeness.

---

> > ### Comment · Reviewer_wwF9 · 2022-08-09
> > **Thanks for the rebuttal**
> >
> > The authors addressed my questions in the rebuttal, I have no further concerns. I support the acceptance of this paper.

---

> > > ### Author Response · Authors · 2022-08-09
> > > **Thank you!**
> > >
> > > We are glad that our response addressed your concerns! And we are highly grateful for your support towards the acceptance of our work!

---

### Official Review · Reviewer_vTCA · 2022-07-11

**Rating:** 7
**Confidence:** 3
**Soundness:** 3 good
**Presentation:** 4 excellent
**Contribution:** 4 excellent

**Summary:**

* This paper considers the problem of learning a structure (DAG) over a set of observed variables with gradient-based methods. The approach designs a new continuous optimization problem capturing the acyclic property of DAGs, which defines the domain of the log-det function to be M-matrices.
* The authors provide a detailed technical description of the approach as well as extensive experiments.
* The authors describe technically that the proposed approach has more well behaved gradients (perhaps indicative of the improved performance empirically) as well as several other theoretical properties of the approach.
* Empirically, the authors provide extensive experiments which demonstrate the efficiency and effectiveness of the proposed approach.

**Questions:**

* Rather than sampling a DAG structure by ordering the nodes in an undirected graph, why not sample DAGs from something like Price's model? I realize that your experiment setup is following earlier work however. I am curious what if any properties we should be aware of for your generative model of dags. How do in-degree / out-degree distributions look like? How do they differ from degree distribution from undirected graph?
* When / why would quality $Q(\cdot)$ make sense / not-make sense for an evaluation measure of the DAGs? Could it be evaluated on held-out data of the same underlying distribution?
* When / would it make sense to evaluate not just wall clock time, but number of samples $n$?
* It seems like hyperparameter tuning could play a large role in the performance of these approaches. Do I understand correctly that previous work has tuned hyperparameters for these same exact datasets in the cases of re-use? Is there clarity on how all hyperparemters were selected? I wonder what you think about evaluating methods using bayesian hyperparameter optimization to find, for each dataset (e.g., setting of generative model hyperparameters) best setting of hyperparameters.
* There is a note about the proposed approach being better at detecting large cycles. For give me if I have missed something, but was there analysis in the experiments section, which noted how much of the improved performance (e.g. SHD) was due to better recovery of such large cycles?
* If these methods were used as part of some downstream approach, do you think the empirical analysis here is predictive of performance? Why/why not / what other measurements would be interesting to consider?

**Limitations:**

Yes, the authors have done a very nice job with this.

**Strengths And Weaknesses:**

Overall I believe that this is a strong paper with an interesting approach for what is becoming a more well studied area (continuous optimization approaches for structure learning).

In particular strengths include:

* **Technical Contribution** - I believe the authors have clearly explained the problem they are solving, why their methodical approach solves the problem, and what are the challenges and insights that are required in the contribution. I find the contribution to be interesting and meaningful.
* **Empirical benefits** - The authors have demonstrated the performance characteristics of their approach with extensive experiments, which would allow a practitioner to consider such results and then determine which methods are most suitable for the application (based on metrics such as time, SHD, TPR, etc). And most often, DAGMA would be a top selection, especially in terms of efficiency compared to methods like NOTEARS.

I have a few questions which indicate places for improvement regarding setup of experiments and measurements, please refer to questions section for detailed description of these weaknesses.

---

> ### Author Response · Authors · 2022-08-02
> **Rebuttal Part 1**
>
> We thank the reviewer for their positive assessment of our work and for their insightful comments and suggestions. We next address your questions in order.
>
> > Rather than sampling a DAG structure by ordering the nodes in an undirected graph, why not sample DAGs from something like Price's model? I realize that your experiment setup is following earlier work, however. I am curious what if any properties we should be aware of for your generative model of dags. How do in-degree / out-degree distributions look like? How do they differ from degree distribution from undirected graph?
>
> Thanks for bringing up this question! In fact, we were imprecise in L640 by saying *"The random graph models above are undirected graphs"*, when in fact **only the ER graphs are undirected graphs and then oriented via a random ordering of the nodes**. For the case of scale-free (SF) graphs, they are directly sampled as directed graphs, i.e., they are DAGs from the start. Interestingly, the "original" Barabasi-Albert (BA) model is precisely the undirected version of the Price’s model, and the Python library we used (igraph) supports generating a directed BA model, which is equivalent to the Price’s model when the exponent in the preferential attachment process is 1. **This means that the scale-free networks we generated follow Price’s model!**
>
> Thanks to your comment, we have flagged this and will make the DAG generation schemes more precise in the camera ready. We will correct L640 and add a note regarding Price’s model.
> Regarding the (undirected) degree distributions, these would follow from known results in ER and BA models. That is, for ER graphs the degree distribution is $P(k)= {n-1 \choose k} p^{k}(1-p)^{n-1-k}$, while for BA graphs the degree distribution is $P(k) \sim k^{-3}$. Regarding the in- and out- distributions, for ER models it is very challenging to characterize this since it is sensitive to the random ordering of the nodes. For BA models, due to the generative process, the in-degree distribution is basically just a shift of the (undirected) degree distribution. For illustration, we computed the empirical in-, out-, and undirected degree distributions for graphs ER1/SF1/ER2/SF2/ER4/SF4 of 50 nodes over 500 repetitions. The figures are contained in this link: https://postimg.cc/gallery/BJ09dHH. It is worth noting that SF graphs have a large number of nodes with in-degree equal to zero (i.e., nodes with no parents) and a few nodes with a large number of parents (i.e., hubs). In contrast, in ER graphs the in- and out- distributions follow a similar pattern.
>
>
> > When / why would quality Q(⋅) make sense / not-make sense for an evaluation measure of the DAGs? Could it be evaluated on held-out data of the same underlying distribution?
>
> This is a good question, and has a surprisingly deep answer. In short, it is well-known that predictive measures such as Q(.) are **not** good measures in the context of structure learning (although we hasten to point out, it could be useful for downstream tasks such as prediction). In fact, it can be proven that predictive metrics are *provably* suboptimal in the sense that they lead to false discoveries; see Meinshausen and Buhlmann (2006) for more details. Since our focus is on structure learning, it does not make sense to use Q for evaluating a DAG (i.e. learning a DAG with the best accuracy possible or equivalently the lowest SHD).
>
> > When / would it make sense to evaluate not just wall clock time, but number of samples n?
>
> While of course the dependence on the number of samples is relevant, recall that our contribution is a new acyclicity regularizer, *which does not depend on the number of samples*. This acyclicity function defines the feasible set (namely DAGs) for the score-based optimization problem, i.e. the feasible set is the same for all methods. Indeed, the only term that depends on $n$ is the score/loss function, for which we are not proposing anything new.

---

> > ### Author Response · Authors · 2022-08-02
> > **Rebuttal Part 2**
> >
> > > It seems like hyperparameter tuning could play a large role in the performance of these approaches. Do I understand correctly that previous work has tuned hyperparameters for these same exact datasets in the cases of re-use? Is there clarity on how all hyperparemters were selected? I wonder what you think about evaluating methods using bayesian hyperparameter optimization to find, for each dataset (e.g., setting of generative model hyperparameters) best setting of hyperparameters.
> >
> > Consistent with previous work in this area (e.g. NOTEARS and its follow-ups), we have not performed any hyperparameter optimization: This is to avoid presenting unintentionally biased results. As a concrete example, in our experiment section, we simply chose a reasonable value for the $\ell_1$ penalty coefficient and used that same value for all ER and SF graphs across many different numbers of nodes. That is to say that one could find even better SHD with proper hyperparameter tuning for each dataset. Therefore, currently, there is no clarity on what the optimal set of hyperparameters for a given dataset is. Finally, we fully agree that Bayesian hyperparameter optimization would be a fascinating approach for this problem to explore in the near future!
> >
> > > There is a note about the proposed approach being better at detecting large cycles. Forgive me if I have missed something, but was there analysis in the experiments section, which noted how much of the improved performance (e.g. SHD) was due to better recovery of such large cycles?
> >
> > Great point! For a direct comparison between acyclicity regularizers, we ran experiments using the *exact same Algorithm 1* and *just replacing the acyclicity regularizer*. We ran simulations on Linear Gaussian SEMs for ER2/ER4/SF2/SF4 graphs. The table below shows the average SHD with 95% confidence intervals on 10 repetitions. DAGMA corresponds to Algorithm 1 using the logdet function, while DAGMA_EXPM corresponds to Algorithm 1 using the trace exponential function from NOTEARS. We observe clearly that our logdet function plays an important role in obtaining DAGs with significantly lower SHD.
> >
> > |method     |graph_type |20         |40           |60           |80           |100           |
> > |:----------|:----------|:----------|:------------|:------------|:------------|:-------------|
> > |DAGMA      |ER2        |0.5 ± 0.61 |1.7 ± 2.06   |1.8 ± 1.15   |0.6 ± 0.9    |4 ± 3.78      |
> > |DAGMA_EXPM |ER2        |2.1 ± 1.61 |10.4 ± 6.81  |16.9 ± 9.3   |20.9 ± 7.67  |33.4 ± 7.44   |
> > |DAGMA      |ER4        |3.8 ± 2.78 |6.5 ± 4      |9.8 ± 5.72   |14.3 ± 6.52  |12.3 ± 5.66   |
> > |DAGMA_EXPM |ER4        |10.7 ± 4.5 |41.4 ± 16.47 |65.5 ± 15.5  |91 ± 27.6    |113.2 ± 27.37 |
> > |DAGMA      |SF2        |0.1 ± 0.23 |0.6 ± 0.77   |0.3 ± 0.48   |1.4 ± 2.47   |1.1 ± 1.09    |
> > |DAGMA_EXPM |SF2        |2 ± 2.51   |2.2 ± 1.99   |4.8 ± 7.28   |1.8 ± 1.58   |7.4 ± 6.94    |
> > |DAGMA      |SF4        |5.9 ± 4.12 |11.2 ± 6.99  |18.6 ± 17.69 |3.5 ± 1.92   |9.9 ± 13.21   |
> > |DAGMA_EXPM |SF4        |8.1 ± 4.84 |17.3 ± 9.73  |28 ± 16.24   |16.4 ± 12.46 |29.9 ± 26.69  |
> >
> > > If these methods were used as part of some downstream approach, do you think the empirical analysis here is predictive of performance? Why/why not / what other measurements would be interesting to consider?
> >
> > Great question! Unfortunately, we cannot make a formal claim. This is a fundamental question with roots in transportability/generalization that makes for exciting future work.

---

> > > ### Comment · Reviewer_vTCA · 2022-08-08
> > > **Thank you for your detailed response.**
> > >
> > > My sincerest apologies for entering this discussion so late. I truly appreciate the authors taking the time and effort to address each of my questions and concerns. Thank you. Reading your response helped clarify me as to the doubts I had.

---

> > > > ### Author Response · Authors · 2022-08-09
> > > > **Thank you!**
> > > >
> > > > We are happy that our response clarified your doubts! And thank you again for your positive assessment of our work!

---

### Official Review · Reviewer_UUUC · 2022-07-11

**Rating:** 7
**Confidence:** 3
**Soundness:** 3 good
**Presentation:** 4 excellent
**Contribution:** 3 good

**Summary:**

The paper introduces a new penalty for enforcing acyclicity of a weighted adjacency matrix for learning DAGs with differentiable optimization. The penalty is zero if and only if the adjacency matrix is acyclic. In contrast to previous proposals based on matrix polynomials, the introduced penalty is based on the log-determinant and is connected to the theory of M-matrices. The paper studies the properties of the new penalty and compares the penalty with previous proposals. From theoretical analysis and empirical observations, it is argued that the new penalty is more suited for learning DAGs / SEMs. Through numerical experiments, it is shown that the new penalty leads to speedups and higher precisions compared to previous proposals.

**Questions:**

1. On discounting long cycles.

Argument (i) claims that the new penalty does not discount a long cycle in the same way some previous proposals do. From Figure 2, it looks persuasive that this could be a big advantage. I would like to see some asymptotic analysis that makes this comparison precise.

2. On vanishing gradients.

Lemma 5 seems insufficient to show that the new penalty does not suffer from vanishing gradients (when the other penalties do). Is there a more compelling argument?

3. On computational speedups.

Despite the same computational complexity, argument (iii) curiously claims that the new penalty "can be computed in about an order of magnitude faster" than the other two in practice. Is there an explanation?

4. Minor issues

(1) line 42: missing power in the expression?

(2) Lemma 1: I would rather state (i) as "DAGs are in the interior of $\mathbb{W}^s$".

**Limitations:**

I do not foresee any potential negative societal impact of their work

**Strengths And Weaknesses:**

### Strengths
1. The paper makes progress on recent developments for differentiable learning of DAGs.
2. The proposed penalty seems novel and has interesting connections to M-matrices.
3. The paper compares the new penalty with previous ones in the literature from various aspects.
4. The paper is very well-written.

### Weaknesses
1. Several aspects in the comparison are based on heuristic arguments or empirical observations, which weakens the claim that the new penalty should be preferred over the earlier ones. A theoretical analysis is desirable to show that the advantage is real or "significant" in typical scenarios or when the problem size grows large.

---

> ### Author Response · Authors · 2022-08-02
> **Rebuttal**
>
> *We thank the reviewer for appreciating the novelty of our contributions and for considering our work to be well-written*. We next address your comments/concerns in order.
>
> ### Strengths And Weaknesses:
>
> Regarding your comment ``several aspects in the comparison are based on heuristic arguments or empirical observations, which weakens the claim that the new penalty should be preferred over the earlier ones’’, the only purely empirical observation is given in Argument (iii) on the observed runtimes, for Arguments (i) and (ii) we provide an explicit justification (Lemmas 4 and 5) for preferring the logdet function over existing ones. As this concern is related to your questions, we elaborate further below.
>
> ### Questions:
>
> Before addressing the questions, we would like to point out that **we provided more details in Section B.2 of the supplement regarding the arguments given in Section 3.2.**
>
> > On discounting long cycles.
>
> * Great question! We shall note that the statement in Lemma 4 follows by a **precise comparison** between $h_{\mathrm{expm}}$ and $h_{\mathrm{logdet}}$ via the spectrum of $W\circ W$ (see its proof in Section A.5). However, following your suggestion, we can provide the stronger asymptotic statement for cycle graphs (as in Fig 2), and we will update Lemma 4 as follows:
>
>     > **Lemma 4.** For all $W \in \mathbb{W}^{s=1}$, we have $h_{\mathrm{poly}}(W) \leq h_{\mathrm{expm}}(W) \leq h_{\mathrm{ldet}}^{s=1}(W)$. Moreover, when $W$ is a cycle graph (as in Figure 2), we have that $h_{\mathrm{expm}}(W) = o(h_{\mathrm{ldet}}(W))$.
>
>     We note two things. First the inequality in Lemma 4 is tight in the sense that, when $W$ is a DAG, all three formulations are exactly equal to zero. Second, *for cycle graphs*, $h_{\mathrm{expm}}(W) = o(h_{\mathrm{ldet}}(W))$ states that while asymptotically both $h_{\mathrm{expm}}(W)$ and $h_{\mathrm{ldet}}(W)$ get to zero, the logdet functions does so  **at a much slower rate** when compared to the expm (and hence also the poly) characterization. In other words, DAGMA penalizes cycles more heavily in a precise way. We corroborate the latter, in the following table where we show the values of the three acyclicity functions for much larger values of $d$. We will elaborate this discussion in the revised version.
>
>     |Function|10|15|50|100|500|1000|2500|5000|7000|
>     |:---:|:---:|:---:|:---:|:---:|:---:|:---:|:---:|:---:|---|
>     | $h_{\mathrm{logdet}}$ | 4.6 | 4.2 | 3.0 | 2.3 | 9.3e-1 | 4.5e-1 | 8.5e-2 | 6.7e-3 | 9.1e-4 |
>     | $h_{\mathrm{expm}}$ | 2.7e-6 | 1.1e-11 | 0 | 0 | 0 | 0 | 0 | 0 | 0 |
>     | $h_{\mathrm{poly}}$ | 1.0e-9 | 0 | 0 | 0 | 0 | 0 | 0 | 0 | 0 |
>
>     We also note that since our method is gradient-based, it is perhaps more important to understand the behavior of the gradient of $h_{\mathrm{ldet}}$, and how it compares to the gradients of existing acyclicity functions, which we discuss next.
>
> > On vanishing gradients.
>
> * Thank you for raising this question! We realize that Lemma 5, as written, is not too informative. We offer to make the following update to Lemma 5:
>
>    > **Lemma 5.** For any walk of length $k$, its contribution to the gradients $\nabla h_{\mathrm{expm}}(W)$ and $\nabla h_{\mathrm{poly}}(W)$ are diminished by $1/k!$ and ${ d-1 \choose k}/(d-1)^k$, respectively. In contrast, $\nabla h_{\mathrm{ldet}}^{s=1}(W)$ does not diminish any walk of any length. This implies that $\left|\nabla h_{\text {poly }}(W)\right| \leq\left|\nabla h_{\operatorname{expm}}(W)\right| \leq\left|\nabla h_{\text {ldet }}^{s=1}(W)\right|$.
>
>     Note that the update above does not require any new proof technique, **the argument is already contained in Section A.6**. Specifically, the above follows by looking at the expansions of the respective gradients of the different acyclicity functions, see the proof of Lemma 5 in Section A.6. These expressions, given in Line 562, clearly show how the gradient of the logdet function weighs all walks/cycles equally irrespective of their lengths, whereas the polynomial and exponential functions diminish walks/cycles of length $k$ by ${ d-1 \choose k} / (d-1)^k$ and $1/k!$, respectively, hence both being prone to vanishing gradients.
>
> > On computational speedups.
>
> * We meant by ``an order of magnitude faster’’ the fact that one can compute the logdet function and its gradient about 10x faster, as stated in Line 280. If the wording creates confusion, we can plainly say it is about 10x faster. Regarding the explanation for this, our best explanation is already given in Section B.2, that is, computing the logdet and its gradient enjoy the large body of work on optimized libraries for matrix factorizations and solving linear systems, while computing the matrix exponential is notoriously tricky, see Lines 618-624 and the references therein.
>
> > Minor issues
>
> (1) Yes, it should be $\mathrm{Tr}(I + \frac{1}{d} W \circ W)^d - d$. Thank you for catching the typo!
>
> (2) Thank you for the suggestion, we will revise this accordingly.

---

> > ### Comment · Reviewer_UUUC · 2022-08-08
> > **Thanks for detailed reply**
> >
> > I would like to thank the authors for addressing my comments in detail.
> >
> > I think the reply has addressed my concerns! I would suggest revising the parts in the paper accordingly so that the quantitative statements can be made where possible.

---

> > > ### Author Response · Authors · 2022-08-09
> > > **Thank you!**
> > >
> > > We are glad to see that our response has clarified your concerns! We hope you feel even more positive about our work. We will make the revision for the camera ready

---

> > > > ### Comment · Reviewer_UUUC · 2022-08-09
> > > > **Update**
> > > >
> > > > I have updated my overall assessment to "7: Accept".

---

### Official Review · Reviewer_kNuE · 2022-07-12

**Rating:** 5
**Confidence:** 5
**Soundness:** 2 fair
**Presentation:** 3 good
**Contribution:** 2 fair

**Summary:**

The authors proposed a new DAG constraints based on the log-determinant of the adjacency matrices. Compare to the original polynomial based DAG constraints, the new DAG constraints do have some good properties and it achieves better performance on ER4 and SF4 graphs with node from 200 to 1000.

**Questions:**

Is the performance from different implementation? In NOTEARS, the authors does not decrease mu but increase the coefficients of h(W dot W). In GOLEM, the authors use fixed coefficient for L1. If put the DAG constraint in the same optimisation framework as NOTEARS, or if we put the NOTEARS in the same framework as the work, what is the performance?

**Ethics Review Area:**

["I don’t know"]

**Limitations:**

It is discussed.

**Strengths And Weaknesses:**

Pros:

1. The proof and derivation of the new DAG constraint is very clear.
2. The illustration of the behaviour of h_det is very good.


Cons:

1. The properties (ii) in Theorem 1 is actually as bad as the Exponential or Polynomial based DAG constraint. See [1,2] for more details. With  this property you will not be able to obtain an DAG unless you let mu in Algorithm 1 converge to zero.

2. Corollary 2 may not be correct and Corollary 1 maybe meaningless. For Corollary 2, it only considers h_det, without consider the score function. By [1,2] without attain the limit point it would not be possible to obtain a DAG, and this will require infinite many steps of optimisation.

3. In the experiment part, the hyper parameter for GOLEM may not be quite right. I have tried GOLEM on SF{1,2,3,4}, and ER{1,2,3}, for numbers of nodes in {10, 20, 50, 100}. The performance of GOLEM performance is consistently better than NOTEARS in a large margin, which is quite different from the result reported in the paper. Also for all cases it would be good if the authors can provide results on graphs with nodes in  {10, 20, 50, 100}. This is because with the same number of expected edges, the more the nodes, the sparser the graph is. Also it would be good if the authors can provide results on ER2, ER3 and SF2, SF3, SF4 with all number of nodes from 10 to 1000. This can provide a fair comparison for different methods in different situations. I have also attached the GOLEM code I have used, you only need to replace the HTorch part with the exponential DAG constraints from NOTEARS.



[1] Wei, Dennis, Tian Gao, and Yue Yu. "DAGs with No Fears: A closer look at continuous optimization for learning Bayesian networks." Advances in Neural Information Processing Systems 33 (2020): 3895-3906.
[2] Ng, Ignavier, et al. "On the convergence of continuous constrained optimization for structure learning." International Conference on Artificial Intelligence and Statistics. PMLR, 2022.


The GOLEM code I have used.


"""

    import numpy as np
    import torch
    import torch.nn as nn
    from .dag import HTorch
    h_torch = HTorch.apply
    class GolemEVModel(nn.Module):
        """
        Set up the objective function of GOLEM.
        Hyperparameters
        (1) GOLEM-NV: equal_variances=False, lambda_1=2e-3, lambda_2=5.0.
        (2) GOLEM-EV: equal_variances=True, lambda_1=2e-2, lambda_2=5.0.
        """
        def __init__(self, d, lambda_1, lambda_2, eps=1e-6, h_type='exponential'):
            """
            Initialize self.
            Parameters
            d: int
                Number of nodes.
            lambda_1: float
                Coefficient of L1 penalty.
            lambda_2: float
                Coefficient of DAG penalty.
            equal_variances: bool
                Whether to assume equal noise variances
                for likelibood objective. Default: True.
            """
            super().__init__()

            self.d = d
            self.lambda_1 = lambda_1
            self.lambda_2 = lambda_2
            self.eps = eps
            self.h_type = h_type
            self._B = nn.Parameter(torch.zeros(self.d, self.d))

        def forward(self, cov_emp):
            # Placeholders and variables
            self.cov_emp = cov_emp
            self.B = self._preprocess(self._B)

            # Likelihood, penalty terms and score
            self.likelihood = self._compute_likelihood()
            self.L1_penalty = self._compute_L1_penalty()
            self.h = self._compute_h()
            self.score = (self.likelihood + self.lambda_1 * self.L1_penalty +
                      self.lambda_2 * self.h)

        def _preprocess(self, B):
            """
            Set the diagonals of B to zero.
            Parameters
            B: tf.Tensor
                [d, d] weighted matrix.

            Return
            torch.Tensor: [d, d] weighted matrix.
            """
            return (1. - torch.eye(self.d)) * B

        def _compute_likelihood(self):
            """
            Compute (negative log) likelihood in the linear Gaussian case.
            Return
            torch.Tensor: Likelihood term (scalar-valued).
            """
            I = torch.eye(self.d)
            return 0.5 * self.d * torch.log(
                torch.trace((I - self.B).T @ self.cov_emp
                        @ (I - self.B))) - torch.linalg.slogdet(I - self.B)[1]

        def _compute_L1_penalty(self):
            """
            Compute L1 penalty.
            Return
            tf.Tensor: L1 penalty term (scalar-valued).
            """
            return torch.norm(self.B, p=1)

        def _compute_h(self):
            """
            Compute DAG penalty.

            Return
            torch.Tensor: DAG penalty term (scalar-valued).
            """
            return h_torch(self.B * self.B, self.h_type, self.eps)
    def golem_ev(X,
             lambda_1=2e-2,
             lambda_2=5.0,
             learning_rate=1e-3,
             num_iter=2e+4,
             graph_thres=0.3,
             eps=1e-6,
             h_type='exponential'):
        n, d = X.shape
        cov_emp = np.cov(X.T, bias=True)
        cov_emp = torch.Tensor(cov_emp)
        model = GolemEVModel(d, lambda_1, lambda_2, eps, h_type)
        train_op = torch.optim.Adam(model.parameters(), lr=learning_rate)

        for i in range(int(num_iter)):
            model(cov_emp)
            score, likelihood, h, B_est = model.score, model.likelihood, model.h, model.B
            loss = score
            train_op.zero_grad()
            loss.backward()
            train_op.step()

        return B_est.detach().numpy()
"""

---

> ### Author Response · Authors · 2022-08-02
> **Our experiments do not contradict GOLEM results. Our contributions also extend to nonlinear models.**
>
> We thank the reviewer for their comments, which we address in order.
>
> ### Summary
>
> In your summary, it is stated that our method "achieves better performance on ER4 and SF4 graphs with nodes from 200 to 1000". While this is correct, **we also provided experiments for small to moderate numbers of nodes** ($d \in [20,100]$) in the supplement, for **both linear and nonlinear models**, see for instance Sections C.1.1, C.1.4, C.2.1, C.2.2. Moreover, as mentioned in Remark 4 in the supplement, we focused mainly on **ER4 and SF4 graphs as they are harder to learn than ER1, ER2, SF1, SF2**. We regret that we did not mention in the main text that we included experiments on these regimes in the supplement. We will add a sentence explaining this in the revision.
>
> ### Strengths And Weaknesses:
>
> About the pros, please note that **an important strength of our method is that it also works for nonlinear models** (see Section C.2), in contrast to **GOLEM which is limited to linear models**.
>
> About the cons:
>
> 1. First, note that in Algorithm 1 we use a constant decay factor for $\mu$. **This is just for practical reasons**, it is of course possible to fix the values of $\mu$ for each iteration, e.g., if we run Algorithm 1 for 5 iterations, we can set $\mu$ to take values in$\\{1, 10^{-1}, 10^{-2},10^{-3}, 0 \\}$. Then, at the last iteration, by the invexity of $h$, the solution will be a DAG.
>
>     Second, **we do not see property (ii) of Theorem 1 as a negative property.** The invexity of $h$ is due to property (ii), and we argue that this is a nice property to have. When $\mu = 0$, we are solving an unconstrained problem where all stationary points are equally good (global min), hence, as pointed out in Remark 3, all we need to have is a good initial point that sits in a basin of attraction where the attractors are close to the ground-truth. As also stated in Remark 3, the score function is used to guide the initial points as $\mu$ gets smaller.
>
>     Third, we also note that in [1] the authors show that a feasible solution (a DAG) does not satisfy the *constraint qualifications* needed for KKT optimality, i.e., a DAG cannot be a stationary point of the Lagrangian nor the Augmented Lagrangian; thus, the authors proceed to reformulate the problem. A similar argument is provided in [2]. Along these lines, let us take a look at the Fritz-John (FJ) condition, which is a **necessary condition for optimality without the need for constraint qualifications**. The FJ condition states that there exists a non-zero vector $\nu = [\nu_0, \nu_1]$ such that:
>             $$\nu_0 \nabla Q(W) + \nu_1 \nabla h(W) = 0,$$
>     where $Q(W)$ denotes the score on W (with or without $\ell_1$ regularization, as this does not affect the argument). Now, when $W$ is a DAG, we have $\nabla h(W) =0$, thus, it must hold that $\nu_0 = 0$ and we can simply set $\nu_1 = 1$. Our Algorithm 1 also resembles this fact, where $\mu$ corresponds to $\nu_0$.
>
> 2. We would appreciate it if the reviewer could elaborate on the correctness of Corollary 2. Corollary 2 indeed only considers $h$ and not the score as we discuss the properties of $h$ in Section 3.1. About requiring "infinite steps to obtain a DAG", see point 1 above. **Corollary 1 is not meaningless whatsoever!** This is yet another nice and useful property from $h_{\mathrm{ldet}}$ that also motivates the design of Algorithm 1. In a few words, whenever $W \in \mathbb{W}^s$, Corollary 1 ensures that the next iterate remains inside $\mathbb{W}^s$ with a suitable step size. This is crucial since as discussed at L185-191, the logdet characterization is only valid in this set.  Finally, there is a typo in Corollary 1, it should read ``...negative gradient…’’, we will correct this in the revision.
>
> 3. **Our results do not contradict the GOLEM results.** Our experiments cover settings not originally studied by GOLEM, so naturally our results are more comprehensive. In the same settings as GOLEM (ER4/SF4 with $d \in [20, 100]$ see Section C.1.1 and Figure 6, as well as, ER2 with $d \in [200, 2000]$ see Section C.1.3 and Figure 8), our results are consistent with their original findings (see Figures 6.(a) and 8 in the GOLEM paper).
>
>     As stated in L687 of the supplement, we use the GOLEM code available at https://github.com/ignavierng/golem, which is the original implementation from the GOLEM authors. In the same paragraph, we also mention that we use the default values of hyperparameters, namely $\lambda_1 = 0.02, \lambda_2=5$, which seems to be the same set of values in your implementation.
>
>     As noted above in our response to your summary comments, we also provided experiments for $d \in [20,100]$. **As shown in Figure 6, GOLEM performs better than NOTEARS for ER4/SF4 graphs for $d \in [20,100]$**, *which does not contradict your observation*. However, **in Figure 9, observe that GOLEM performs worse than NOTEARS for ER6 graphs in the same regime of $d \in [20,100]$**.
>
>     *Our response continues below.*

---

> > ### Author Response · Authors · 2022-08-02
> > **Part 2 of rebuttal**
> >
> > 3. Also, in the GOLEM paper, the authors only experimented on ER2 graphs for the high dimensional setting (Figure 8 in the GOLEM paper). We replicated their experiments in Section C.1.3. We experimented with ER2 graphs for $d \in [200, 2000]$, and as seen in Figure 8, GOLEM improves its performance in this sparser setting w.r.t. the ER4/SF4 graphs shown in Figure 7. Nonetheless, **DAGMA still attains better SHD in both ER2 and ER4 graphs**.
> >
> >     At the request of the reviewer, for completeness we next show experiments on sparser linear Gaussian SEMs such as ER{1,2,3} and SF{1,2,3} for a number of nodes in $\\{20, 40, 60, 80, 100, 200, 400, 600\\}$. The table below shows the average SHD with 95% confidence intervals on 10 repetitions. We note that, not surprisingly, DAGMA and GOLEM perform similarly in ER1/SF1, with DAGMA already taking the lead in ER2 graphs. We also point out that NOTEARS already performs better than GOLEM in ER3/SF3 graphs for $d>200$.
> >
> >     If the reviewer still has concerns regarding the comparison of GOLEM against NOTEARS, it would be helpful if the reviewer could report their results explicitly and specify the setting used for NOTEARS. It is worth noting that in the GOLEM paper, the authors used a $\ell_1$ penalty coefficient of 0.1 for NOTEARS which we argue that it was not a fair comparison to use very different levels of sparsity, in the table below all three methods used the same $\ell_1$ coefficient of 0.03.
> >
> > |Method|Graph|20|40|60|80|100|200|400 |600|
> > |:-------:|:----------:|:-------:|:---------:|:---------:|:---------:|:---------:|:----------:|:-----------:|:-----------:|
> > |DAGMA|ER1|0.1±0.2|0.3±0.7|0.9±0.9|1±1.6|1.5±1.2|0.2±0.4|3.7±2.1|3.6±1.8|
> > |GOLEM|ER1|0±0|0.3±0.5|0.3±0.7|0.2±0.4|0.6±0.8|1.9±2.6|1.3±1|1.9±1.4|
> > |NOTEARS|ER1|0.5±0.9|0.5±0.7|1.7±1.1|1.7±1.8|2.4±1.8|8.3±6.2|12.6±2.6|14.9±7.1|
> > |DAGMA|ER2|0.5±0.6|1.7±2.1|1.8±1.1|0.6±0.9|4±3.8|7.3±4|14.9±9|20.9±8.5|
> > |GOLEM|ER2|1.4±1.4|1.4±1.2|2.3±1.8|5.1±3.2|7.5±5|13.2±6.1|44.5±16.4|68.6±16.2|
> > |NOTEARS|ER2|2.3±1.6|7.9±5.9|11.1±8.9|17.7±10.9|27.4±12.2|49.4±18.8|108.4±29.2|163.8±33.2|
> > |DAGMA|ER3|2±2.3|3.3±3.1|3.3±2.6|6.3±3.4|7±4.6|19.9±14.4|58.8±29.6|109.9±26.1|
> > |GOLEM|ER3|6±4.9|6.4±8.6|8.4±3.2|18.4±7.9|33±18.1|77.4±25.9|334.4±223.1|859.3±312.2|
> > |NOTEARS|ER3|4.1±2.5|14±9.3|37.2±15.9|37.2±13.9|55.5±18.4|114.8±36.9|285.6±52.6|455.1±72.8|
> > |DAGMA|SF1|0±0|0±0|0±0|0±0|0.4±0.6|0.6±1.4|1.5±2|3.1±5.6|
> > |GOLEM|SF1|0.1±0.2|0.1±0.2|0±0|0±0|0.2±0.4|0.5±0.5|0.3±0.3|1±1.1|
> > |NOTEARS|SF1|0±0|0±0|0.1±0.2|0.1±0.2|0.1±0.2|0±0|1.5±2|4.2±8.5|
> > |DAGMA|SF2|0.1±0.2|0.6±0.8|0.3±0.5|1.4±2.5|1.1±1.1|0.3±0.5|1.3±1.4|2.2±1.2|
> > |GOLEM|SF2|0.4±0.9|0.7±1.1|3.2±5.4|0.3±0.5|1.3±1.4|2.4±3.6|5.4±5.6|44.3±59.3|
> > |NOTEARS|SF2|0.5±0.6|1.3±1.8|2.6±3.8|1±0.7|2.3±2.9|4.3±3.7|5.9±4.6|15.5±13|
> > |DAGMA|SF3|2.7±3.3|2.5±4.2|8.1±11.4|2.8±3.4|15.9±21.4|7.6±8.1|5.8±3.5|13.6±10.2|
> > |GOLEM|SF3|1.3±1.2|1.7±1.7|1.5±2.5|5.4±6|7.3±6.5|9±7.1|68.6±55.4|510.4±414.2|
> > |NOTEARS|SF3|2.7±3.2|8.4±7.3|7.1±8.9|5.1±2.3|12.9±15.9|10.9±6.9|19.3±10.3|68.4±61.8|

---

> > > ### Author Response · Authors · 2022-08-02
> > > **Part 3 of the rebuttal**
> > >
> > > ### Questions
> > > > ``Is the performance from different implementation? In NOTEARS, the authors does not decrease mu but increase the coefficients of h(W dot W). In GOLEM, the authors use fixed coefficient for L1.’’
> > >
> > > We used the authors’ implementations: As explained above and in Section C.1., we use the original GOLEM implementation provided at https://github.com/ignavierng/golem. For NOTEARS, we used the code available at https://github.com/xunzheng/notears.
> > >
> > >
> > > >``If put the DAG constraint in the same optimisation framework as NOTEARS, or if we put the NOTEARS in the same framework as the work, what is the performance?’’
> > >
> > > Thank you for the insightful question!
> > > * We can indeed use the Augmented Lagrangian scheme using our logdet characterization. However, part of our contributions (namely Algorithm 1) is to argue that such a scheme or the quadratic penalty method are not necessary for obtaining more accurate structures *using the logdet formulation*. Implementation-wise, simply replacing the trace exponential by the logdet function in the NOTEARS code will not necessarily work since NOTEARS uses the scipy minimize function, which does not take into account the fact that we need to stay in the interior of $\mathbb{W}^s$.
> > >
> > > * We can, however, replace our logdet function by the trace exponential in Algorithm 1, the performance is reported in the table below. We ran experiments on Linear Gaussian SEMs for ER2/ER4/SF2/SF4 graphs. DAGMA_EXPM refers to Algorithm 1 using the trace exponential function from NOTEARS instead of logdet. We thank the reviewer for bringing up this question as  this a clear example of the benefits of the logdet formulation when directly compared to the trace exponential function under the same scheme. We will add these experiments in the supplement.
> > > |Method     |graph |20         |40           |60           |80           |100           |
> > > |:----------:|:----------:|:----------:|:------------:|:------------:|:------------:|:-------------:|
> > > |DAGMA      |ER2        |0.5 ± 0.61 |1.7 ± 2.06   |1.8 ± 1.15   |0.6 ± 0.9    |4 ± 3.78      |
> > > |DAGMA_EXPM |ER2        |2.1 ± 1.61 |10.4 ± 6.81  |16.9 ± 9.3   |20.9 ± 7.67  |33.4 ± 7.44   |
> > > |DAGMA      |ER4        |3.8 ± 2.78 |6.5 ± 4      |9.8 ± 5.72   |14.3 ± 6.52  |12.3 ± 5.66   |
> > > |DAGMA_EXPM |ER4        |10.7 ± 4.5 |41.4 ± 16.47 |65.5 ± 15.5  |91 ± 27.6    |113.2 ± 27.37 |
> > > |DAGMA      |SF2        |0.1 ± 0.23 |0.6 ± 0.77   |0.3 ± 0.48   |1.4 ± 2.47   |1.1 ± 1.09    |
> > > |DAGMA_EXPM |SF2        |2 ± 2.51   |2.2 ± 1.99   |4.8 ± 7.28   |1.8 ± 1.58   |7.4 ± 6.94    |
> > > |DAGMA      |SF4        |5.9 ± 4.12 |11.2 ± 6.99  |18.6 ± 17.69 |3.5 ± 1.92   |9.9 ± 13.21   |
> > > |DAGMA_EXPM |SF4        |8.1 ± 4.84 |17.3 ± 9.73  |28 ± 16.24   |16.4 ± 12.46 |29.9 ± 26.69  |

---

> > ### Comment · Reviewer_kNuE · 2022-08-03
> > **About property (ii) of Theorem 1**
> >
> > It is true that you can set the series $\mu$ to {1, 0.1, 0.01, 0.001, 0}. However it will cause a problem.In the last iteration of the algorithm, the objective only include the term that encourages the graph to be DAG, and thus arbitrary DAG is the solution. In this case, a meaningless DAG is obtained in finite steps. Once you want the DAG to be meaningful, it has to take infinite steps.
> >
> > For the FJ condition, it is equivalent to KKT condition if $v_0\neq 0$. However if $v_0 =0$, it is just the Mangasarian–Fromovitz constraint qualification, which is much weaker than the KKT condition. In this case the proposed DAG constraints must suffer from property (ii) of Theorem 1.

---

> > > ### Author Response · Authors · 2022-08-04
> > > **Algorithm 1 **does not** return a "random" DAG if $\mu$ is set to zero in the last iteration**
> > >
> > > Thank you for taking the time to respond to our comments.
> > >
> > > Do you have any comments regarding items (2) and (3)? We have put a lot of effort into running the additional experiments, and *we would appreciate it if you could let us know if it has clarified your concerns*. You had also mentioned that you found an issue with Corollary 2: *Can you please provide more details on this?*
> > >
> > > Regarding the last iteration when $\mu=0$: it will **not be an arbitrary DAG**. This is due to our use of the central path in Algorithm 1, which ensures the result is not arbitrary. Indeed, if setting $\mu=0$ returned an arbitrary DAG, our results would be random instead of close to the ground truth. **We have emphasized our central path approach several times in the paper**, see L17 in the Abstract, L71-72 and L103 in the Introduction, L293, L296, and $\mu$ being the central path coefficient in Algorithm 1 in Section 4.
> > >
> > > We have provided a more detailed explanation of this in parts (1) and (2) of our response above; we would appreciate it if the reviewer could provide a response to our specific points there. (Note that the point about the FJ condition is meant only to provide some intuition behind our approach compared to existing approaches, and is not a rigorous argument.) _To help illustrate the central path more clearly, please see the following example (**https://postimg.cc/75cnskpb**) of a two-node graph **following Figure 1 in the paper**. In this case, the ground truth is the DAG with $w_1=1.2$ and $w_2=0$ with standard Gaussian noises. The title of the plot shows the value of $\mu$, the initial point (red point in the plot), and the final point after performing gradient descent (the cyan point in the plot). The example clearly illustrates the central path behavior of our algorithm, and certainly does not return a "random" DAG._
> > >
> > > Finally, although property (ii) of Theorem 1 suggests LICQ does not hold, LICQ is a necessary qualification for KKT to be a necessary condition, so *KKT is not a necessary condition in this case*. Regardless, **our logdet formulation is still a valid characterization of acyclicity, and it is clear that a similar reformulation as given in NOFEARS will work for our formulation too.** Since the KKT re-formulation is clearly documented in NOFEARS, we opted not to repeat these details in our submission. Nonetheless, we are happy to add a remark and pointer to the NOFEARS paper on this.

---

> > > > ### Comment · Reviewer_kNuE · 2022-08-08
> > > > **Concern on $\mu=0$, as well as experiments.**
> > > >
> > > > Sorry for the late reply. I would like to thank the authors for the informative response.
> > > > I was trying to do some experiments in the paper.
> > > >
> > > > I understand that the paper are using the central path algorithm. However, the limit point of the central path algorithm will be the solution, and it may require infinite number of steps.
> > > >
> > > > If you would like to use the central path algorithm for finite steps, you will have to provide a prove some properties as follows:
> > > >
> > > > + From a sufficient small $\mu$, you get a candidate graph
> > > > +  Use the candidate graph as initial solution, solve the optimization problem in line 3 of algorithm 1 with $\mu=0$ to obtain the final solution.
> > > > +  Show that the distance between the final solution and the candidate graph is bounded.
> > > >
> > > > Without doing this you can not say you will obtain a meaningful DAG in finite steps. This is because arbitrary DAG is a solution of  the optimization problem in line 3 of algorithm 1 with $\mu=0$.
> > > >
> > > >
> > > > Also recently I have tried to use GOLEM on some of the experiments in the paper. My finding is as follows:
> > > >
> > > > + using the hyper parameter provided in the paper, GOLEM achieves similar performances as reported in the paper.
> > > > + By setting learning rate to $2\times 10^{-4}$, and number of iterations to more than $2\times 10^5$, GOLEM achieves similar performance as the DAGMA on 500 node ER4 graphs, but the running time for GOLEM than is more than 2 hours on a 3090 graphics card. Here it is slow because I use double precision and the double precision on 3090 is very slow. I do not know if the authors are using single precision or double precision. If it is single precision, I believe by converting to double precision the performance can be improved.
> > > >
> > > > Why I am trying to do this experiment is because the GOLEM algorithm are specifically designed for sparse graphs. For very large graphs with 500 or 1000 nodes, since the number of degree for each node is fixed, larger number of nodes results in sparser graphs. In this case it would be hard to believe GOLEM would be perform worse than NOTEARS on large scale problems.
> > > >
> > > > The experiments suggests that if we only consider the accuracy of DAG recovery, GOLEM may have similar performance as the proposed methods, but it requires more running time, and the larger the graph is, the slower GOLEM is.

---

> > > > > ### Author Response · Authors · 2022-08-09
> > > > > **Response**
> > > > >
> > > > > We thank the reviewer for clarifying their questions and for taking the time to run additional experiments.
> > > > >
> > > > > **Regarding $\mu=0$:**
> > > > >
> > > > > In fact, as you suggest, **we can prove that the distance between the final solution at $\mu=0$ and the candidate graph is bounded** (see below for details).
> > > > >
> > > > > **Regarding GOLEM:**
> > > > >
> > > > > Thank you for flagging single vs double precision in GOLEM. Following your suggestion, we re-implemented GOLEM under double precision, but unfortunately **have not been able to replicate your claims**. In order to properly evaluate your claims, we feel it is necessary to see your results, including means and standard errors for each run and setting tested. (See below for a summary of our experiments.)
> > > > >
> > > > > Nevertheless, we will update our figures to include GOLEM with double precision, and add a note regarding this detail.
> > > > >
> > > > > ## More Details:
> > > > > ### **Proof sketch for $\mu=0$:**
> > > > > Below we prove the bound requested by the reviewer.
> > > > >
> > > > > 1. Pick $\mu’>0$ and let $W’$ be a solution to $\min_W \mu’ Q(W) + h(W)$. Stationarity implies $\mu’ \nabla Q(W’) + \nabla h(W’) = 0$. Let $L = || \nabla Q(W’)||$, then $|| \nabla h(W’) || = \mu’ L$
> > > > >
> > > > > 2. Let $B$ be the local Lipschitz constant of $\nabla h$ at $W’$. Suppose now that next we solve Line 3 of Algorithm 1 with $\mu = 0$, and we solve it by gradient descent **starting at $W’$**. Then, a short calculation shows that the $t$-th GD iteration is given by: $W^{(t)} = W^{(t-1)} - \eta \nabla h(W^{(t-1)})$. Thus $|| W^{(t)} - W^{(t-1)} || \leq \eta \mu’ L (\eta B +1)^{t-1}$.
> > > > >
> > > > > 3. Summing and telescoping we obtain $|| W^{(t)} - W’ || \leq \eta \mu’ L \sum_{k=0}^{t-1} (\eta B +1)^k$.
> > > > >
> > > > > **It follows that for small $\mu’>0$, the distance between the final solution at $\mu=0$ and the candidate graph for $\mu’$ is bounded, as desired.** As a final remark, note that since $h$ is smooth, by the GD lemma, GD will find a stationary point which is a global min of $h$ by invexity in $O(1/\epsilon)$ iterations, for an $\epsilon$ suboptimality error.
> > > > >
> > > > >
> > > > > ### **Additional experiments:**
> > > > >
> > > > > We compared GOLEM to NOTEARS and DAGMA under the reviewer's set of hyperparameters with double precision enabled. See this figure for the results: https://postimg.cc/RJv8YWvh. For reproducibility, we also report the seeds used to generate the graphs and run GOLEM, $[3223,  189,  860, 1256, 3239, 2727, 4178, 1108, 4361, 4486]$. While there is an improvement in GOLEM’s performance, GOLEM with double precision (GOLEM_DOUBLE in the figure) still performs worse in SHD than NOTEARS and DAGMA for ER4/SF4 and $d \in \\{ 200, 300, 500 \\}$. The runtime is much slower as well, being now in the same order as NOTEARS. Finally, although we used the set of hyperparameters you suggested, unfortunately, we did not observe that GOLEM performs similarly to DAGMA.

---

### Author Response · Authors · 2022-08-02
**To all reviewers**

We thank all reviewers for putting the time to read our work and for providing insightful comments and suggestions to improve our work. We are thrilled to see that in general there is a positive assessment of our contributions and that there is a consensus on the good quality of the presentation of our results. We have addressed your questions and hopefully our responses will help clarify them. We encourage the reviewers to ask any further clarification questions during the discussion week.

---

### Meta-Review · Area_Chair_aywV · 2022-08-24

**Recommendation:** Accept
**Confidence:** Certain

**Metareview:**

Overall, reviews for this paper are quite positive. The paper presents an interesting and effective new approach to incorporating a DAG constraint into an optimization problem by using a characterization of DAGs in terms of the logdet function.

During discussion, the reviewers raised several important questions/points for clarification, which the authors largely addressed in their responses. I encourage the authors to use these responses to guide editing of the paper for the final version.

There was some disagreement during author-reviewer discussion in regards to a comparison to GOLEM made by one of the reviewers. After the discussion period, the reviewer has provided some useful information on possible reasons for inconsistencies. I hope that the authors will investigate these points carefully and update empirical results/discussions as needed in the final version.

From the reviewer:

I carefully compared my version of GOLEM with the version of GOLEM used by the author (released with the GOLEM paper), and there are several differences.
	1	My version is implemented in PyTorch, theirs is implemented in TensorFlow,
	2	In my version, a learning rate scheduler is used to apply smaller and smaller learning rates to solve the problem as it approaches a local optimum, and in theirs, a fixed learning rate is used.

Considering this, I think it is reasonable that the authors did not observe the same performance as I did. Here the learning rate scheduler might play an important role since I also observed that with some large learning rate GOLEM may not converge so that a fixed learning rate may finally converge to a bad solution, or fail to converge.
…
I think the paper can be further enhanced if the authors can replace the DAG constraint in GOLEM with theirs to obtain a new algorithm. From my experience, it is highly possible that with a proper optimization algorithm it can achieve far better performance than the current version.

**Award:**

No

---

### Decision · Program_Chairs · 2022-09-14

Accept